# ORDER-PRESERVING GFLOWNETS

**Yihang Chen**
Section of Communication Systems
EPFL, Switzerland
yihang.chen@epfl.ch

**Lukas Mauch**
Sony Europe B.V.
Stuttgart Laboratory 1, Germany
lukas.mauch@sony.com

## ABSTRACT

Generative Flow Networks (GFlowNets) have been introduced as a method to sample a diverse set of candidates with probabilities proportional to a given reward. However, GFlowNets can only be used with a predefined scalar reward, which can be either computationally expensive or not directly accessible, in the case of multi-objective optimization (MOO) tasks for example. Moreover, to prioritize identifying high-reward candidates, the conventional practice is to raise the reward to a higher exponent, the optimal choice of which may vary across different environments. To address these issues, we propose Order-Preserving GFlowNets (OP-GFNs), which sample with probabilities in proportion to a learned reward function that is consistent with a provided (partial) order on the candidates, thus eliminating the need for an explicit formulation of the reward function. We theoretically prove that the training process of OP-GFNs gradually sparsifies the learned reward landscape in single-objective maximization tasks. The sparsification concentrates on candidates of a higher hierarchy in the ordering, ensuring exploration at the beginning and exploitation towards the end of the training. We demonstrate OP-GFN's state-of-the-art performance in single-objective maximization (totally ordered) and multi-objective Pareto front approximation (partially ordered) tasks, including synthetic datasets, molecule generation, and neural architecture search. [1]

## 1 INTRODUCTION

Generative Flow Networks (GFlowNets) are a novel class of generative machine learning models. As introduced by Bengio et al. (2021b), these models can generate composite objects $x \in \mathcal{X}$ with probabilities that are proportional to a given reward function, i.e., $p(x) \propto R(x)$. Notably, GFlowNets can represent distributions over composite objects such as sets and graphs. Their applications include the design of biological structures (Bengio et al., 2021a; Jain et al., 2022), Bayesian structure learning (Deleu et al., 2022; Nishikawa-Toomey et al., 2022), robust scheduling (Zhang et al., 2023a), and graph combinatorial problems (Zhang et al., 2023b). GFlowNets can also provide a route to unify many generative models (Zhang et al., 2022), including hierarchical VAEs (Ranganath et al., 2016), normalizing flows (Dinh et al., 2014), and diffusion models (Song et al., 2021).

We investigate GFlowNets for combinatorial stochastic optimization of black-box functions. More specifically, we want to maximize a set of $D$ objectives over $\mathcal{X}$, $\boldsymbol{u}(x) \in \mathbb{R}^D$. In the objective space $\mathbb{R}^D$, we define the the *Pareto dominance* on vectors $\boldsymbol{u}, \boldsymbol{u}' \in \mathbb{R}^D$, such that $\boldsymbol{u} \preceq \boldsymbol{u}' \Leftrightarrow \forall k, u_k \leq u'_k$. We remark that $\preceq$ induces a total order on $\mathcal{X}$ for $D = 1$, and a partial order for $D > 1$. [2]

We identify two problems:

1. GFlowNets require an explicit formulation of a scalar reward $R(x)$ that measures the global quality of an object $x$. In the multi-objective optimization where $D > 1$, GFlowNets cannot be directly applied and $\boldsymbol{u}(x)$ has to be scalarized in prior (Jain et al., 2023; Roy et al., 2023). Besides, evaluating the exact value of the objective function might be costly. For example, in the neural architecture search (NAS) task (Dong et al., 2021), evaluating the architecture's test accuracy requires training the network to completion.

---

[1] Our codes are available at https://github.com/yhangchen/OP-GFN.

[2] A partial order is a homogeneous binary relation that is reflexive, transitive, and antisymmetric. A total order is a partial order in which any two elements are comparable.

2. For a scalar objective function $u(x)$ where $D = 1$, to prioritize the identification of candidates with high $u(x)$ value, GFlowNets typically operate on the exponentially scaled reward $R(x) = (u(x))^\beta$, termed as GFN-$\beta$ (Bengio et al., 2021a). Note, that the optimal $\beta$ heavily depends on the geometric landscape of $u(x)$. A small $\beta$ may hinder the exploitation of the maximal reward candidates since even a perfectly fitted GFlowNet will still have a high probability of sampling outside the areas that maximize $u(x)$. On the contrary, a large $\beta$ may hinder the exploration since the sampler will encounter a highly sparse reward landscape in the early training stages.

In this work, we propose a novel training criterion for GFlowNets, termed Order-Preserving (OP) GFlowNets. The main difference to standard GFlowNets is that OP-GFNs are not trained to sample proportional to an explicitly defined reward $R : \mathcal{X} \to \mathbb{R}^+$, but proportional to a learned reward that should be compatible with a provided (partial) order over $\mathcal{X}$ induced by $\boldsymbol{u}$.

We summarize our contributions in the following:

1. We propose the OP-GFNs for both the single-objective maximization and multi-objective Pareto approximation, which require **only** the (partial-)ordering relations among candidates.

2. We empirically evaluate our method on synthesis environment HyperGrid (Bengio et al., 2021a), and two real-world applications: NATS-Bench (Dong et al., 2021), and molecular designs (Shen et al., 2023; Jain et al., 2023) to demonstrate its advantages in the diversity and the top reward (or the closeness to the Pareto front) of the generated candidates.

3. In the single-objective maximization problem, we theoretically prove that the learned reward is a piece-wise geometric progression with respect to the ranking by the objective function, and can efficiently assign high credit to the important substructures of the composite object.

4. We show that the learned order-preserving reward will balance the exploration in the early stages and the exploitation in the later stages of the training, by gradually sparsifying the reward function during the training.

## 2 PRELIMINARIES

### 2.1 GFLOWNET

We give some essential definitions, following Section 3 of (Bengio et al., 2021b). For a directed acyclic graph $\mathcal{G} = (\mathcal{S}, \mathcal{A})$ with state space $\mathcal{S}$ and action space $\mathcal{A}$, we call the vertices *states*, the edges *actions*. Let $s_0 \in \mathcal{S}$ be the *initial state*, i.e., the only state with no incoming edges; and $\mathcal{X}$ the set of *terminal states*, i.e., the states with no outgoing edges. We note that Bengio et al. (2021a) allows terminal states with outgoing edges. Such difference can be dealt with by augmenting every such state $s$ by a new terminal state $s^\top$ with the *terminal action* $s \to s^\top$.

A *complete trajectory* is a sequence of transitions $\tau = (s_0 \to s_1 \to \ldots \to s_n)$ going from the initial state $s_0$ to a terminal state $s_n$ with $(s_t \to s_{t+1}) \in \mathcal{A}$ for all $0 \leq t \leq n - 1$. Let $\mathcal{T}$ be the set of complete trajectories. A *trajectory flow* is a nonnegative function $F : \mathcal{T} \to \mathbb{R}_{\geq 0}$. For any state $s$, define the state flow $F(s) = \sum_{s \in \tau} F(\tau)$, and, for any edge $s \to s'$, the edge flow $F(s \to s') = \sum_{\tau = (\ldots \to s \to s' \to \ldots)} F(\tau)$. The forward transition $P_F$ and backward transition probability are defined as $P_F(s'|s) := F(s \to s')/F(s), P_B(s|s') = F(s \to s')/F(s')$ for the consecutive state $s, s'$. We define the normalizing factor $Z = \sum_{x \in \mathcal{X}} F(x)$. Suppose that a nontrivial nonnegative reward function $R : \mathcal{X} \to \mathbb{R}_{\geq 0}$ is given on the set of terminal states. GFlowNets (Bengio et al., 2021a) aim to approximate a Markovian flow $F(\cdot)$ on the graph $\mathcal{G}$ such that $F(x) = R(x), \forall x \in \mathcal{X}$.

In general, an objective optimizing for this equality cannot be minimized directly because $F(x)$ is a sum over all trajectories leading to $x$. Previously, several objectives, such as *flow matching*, *detailed balance*, *trajectory balance*, and *subtrajectory balance*, have previously been proposed. We list their names, parametrizations and respective constraints in Table 2.1. By Proposition 10 in Bengio et al. (2021b) for flow matching, Proposition 6 of Bengio et al. (2021b) for detailed balance, and Proposition 1 of Malkin et al. (2022) for trajectory balance, if the training policy has full support and respective constraints are reached, the GFlowNet sampler does sample from the target distribution.

### 2.2 MULTI-OBJECTIVE OPTIMIZATION

The Multi-Objective Optimization (MOO) problem can be described as the desire to maximize a set of $D > 1$ objectives over $\mathcal{X}$. We summarize these objectives in the vector-valued function

| Method | Parametrization (by $\theta$) | Constraint |
|---|---|---|
| Flow matching (FM, Bengio et al. (2021a)) | $F(s \to s')$ | $\sum_{(s'' \to s) \in \mathcal{A}} F(s'' \to s) = \sum_{(s \to s') \in \mathcal{A}} F(s \to s').$ |
| Detailed balance (DB, Bengio et al. (2021b)) | $F(s), P_F(s'\|s), P_B(s\|s')$ | $F(s)P_F(s'\|s) = F(s')P_B(s\|s').$ |
| Trajectory balance (TB, Malkin et al. (2022)) | $P_F(s'\|s), P_B(s\|s'), Z$ | $Z \prod_{t=1}^{n} P_F(s_t\|s_{t-1}) = R(x) \prod_{t=1}^{n} P_B(s_{t-1}\|s_t)$ |
| SubTrajectory balance (subTB, Madan et al. (2022)) | $F(s), P_F(s'\|s), P_B(s\|s')$ | $F(s_{m_1}) \prod_{t=m_1+1}^{m_2} P_F(s_t\|s_{t-1}) = F(s_{m_2}) \prod_{t=m_1+1}^{m_2} P_B(s_{t-1}\|s_t)$ |

Table 2.1: GFlowNet objectives. We define the complete trajectory $\tau = (s_0 \to s_1 \to \ldots \to s_n = x)$, and its subtrajectory $s_{m_1} \to \cdots \to s_{m_2}, 0 \le m_1 < m_2 \le n$.

$\boldsymbol{u}(x) \in \mathbb{R}^D$. The Pareto front of the objective set $S$ is defined by $\text{Pareto}(S) := \{\boldsymbol{u} \in S : \nexists \boldsymbol{u}' \neq \boldsymbol{u} \in S, \text{s.t. } \boldsymbol{u} \preceq \boldsymbol{u}'\}$, and the Pareto front of sample set $X$ is defined by $\text{Pareto}(X) := \{x \in X : \nexists x' \neq x, \text{s.t. } \boldsymbol{u}(x) \preceq \boldsymbol{u}(x')\}$. The image of $\boldsymbol{u}(x)$ is $\mathcal{U} := \{\boldsymbol{u}(x) : x \in \mathcal{X}\}$. Such MOO problems are typically solved by scalarization methods, i.e. preference or goal conditioning.

A GFN-based approach to multi-objective optimization (Jain et al., 2023), called *preference-conditioning* (PC-GFNs), amounts to scalarizing the objective function by using a set of preferences $\boldsymbol{w}$: $R_{\boldsymbol{w}}(x) := \boldsymbol{w}^\top \boldsymbol{u}(x)$, where $\mathbb{1}^\top \boldsymbol{w} = 1, w_k \ge 0$. However, the problem of this scalarization is that, only when the Pareto front is convex, we can obtain the equivalence $\exists \boldsymbol{w}, x^\star \in \arg\max R_{\boldsymbol{w}}(x) \Leftrightarrow x^\star \in \text{Pareto}(\mathcal{X})$. In particular, the "$\Leftarrow$" relation does not hold on the non-convex Pareto front, which means that some Pareto front solutions never maximize any linear scalarization. Recently, Roy et al. (2023) proposed *goal-conditioning* (GC-GFNs), a more controllable conditional model that can uniformly explore solutions along the entire Pareto front under challenging objective landscapes. A sample $x$ is defined in the *focus region* $g$ if the cosine similarity between its objective vector $\boldsymbol{u}$ and the goal direction $\boldsymbol{d}_g$ is above the threshold $c_g$, i.e. $g := \{\boldsymbol{u} \in \mathbb{R}^k : \cos\langle \boldsymbol{u}, \boldsymbol{d}_g \rangle := \frac{\boldsymbol{u} \cdot \boldsymbol{d}_g}{\|\boldsymbol{u}\| \cdot \|\boldsymbol{d}_g\|} \ge c_g\}$. The reward function $R_g$ depends on the current goal $g$ so that the conditioned reward will only be non-zero in the focus region, i.e., $R_g(x) := \mathbb{1}^\top \boldsymbol{u}(x) \cdot \mathbf{1}[\boldsymbol{u}(x) \in g]$, where $\mathbf{1}[\cdot]$ is the indicator function.

## 3 METHOD

### 3.1 OVERVIEW

We define some terminology here. Let $T_B := \{(s_0^i \to \cdots \to s_{n_i}^i = x_i)\}_{i=1}^B$ be a batch of trajectories of size $B$, $X_B = \{x_i\}_{i=1}^B$ be the set of terminal states that are reached by the trajectories in batch $T_B$, and $S_B = \{\boldsymbol{u}(x_i)\}_{i=1}^B$ be the (vector) objective set. We use $\boldsymbol{u}(x)$ to denote the single or multi-objective function for a unified discussion, and $u(x)$ for single-objective function only. We will drop the subscript $B$ when the batch size is irrelevant. Let $R(x)$ be the predefined reward function used in previous GFlowNets methods, such as $R(x) = (u(x))^\beta$ when $D = 1$. We call $\beta$ the reward exponent and assume $\beta = 1$ if not mentioned otherwise. We also let $\widehat{R}(x)$ be an undetermined reward function that we will learn in the training.

In the GFlowNet framework, $\widehat{R}(\cdot)$ is parametrized by the GFlowNet parameters $\theta$. The training of the order-preserving GFlowNet can be divided into two parts: 1) The order-preserving criterion, and 2) The Markov-Decision-Process (MDP) constraints.

**1) Order preserving.** We define a local labeling $y(\cdot; X)$ for $x \in X$ on a batch of terminal states (candidates) $X$. Let $\text{Pareto}(X)$ be the non-dominated candidates in $X$, the labeling is defined by $y(x; X) := \mathbf{1}[x \in \text{Pareto}(X)]$, which induces the labeling distribution $\mathbb{P}_y(x|X)$. The reward $\widehat{R}(\cdot)$ also induces a local distribution on the sample set $X$, denoted by $\mathbb{P}(x|X, \widehat{R})$. We have

$$\mathbb{P}_y(x|X) := \frac{\mathbf{1}[x \in \text{Pareto}(X)]}{|\text{Pareto}(X)|}, \quad \mathbb{P}(x|X, \widehat{R}) := \frac{\widehat{R}(x)}{\sum_{x' \in X} \widehat{R}(x')}, \forall x \in X. \tag{1}$$

Since we want $\widehat{R}(\cdot)$ to keep the ordering of $\boldsymbol{u}$, we minimize the KL divergence. The order-preserving loss is, therefore,

$$\mathcal{L}_{\text{OP}}(X; \widehat{R}) := \text{KL}(\mathbb{P}_y(\cdot|X) \| \mathbb{P}(\cdot|X, \widehat{R})). \tag{2}$$

In the TB parametrization, the state flow $F(\cdot)$ is not parametrized. By the trajectory balance equality constraints in Table 2.1, the order-preserving reward of $x$ on trajectory $\tau$ ending at $x$ is therefore

$$\widehat{R}_{\mathrm{TB}}(x;\theta) := Z_\theta \prod_{t=1}^{n} P_F(s_t|s_{t-1};\theta)/P_B(s_{t-1}|s_t;\theta). \tag{3}$$

For the non-TB parametrizations, where the state flow is parametrized, we let $\widehat{R}(x;\theta) = F(x;\theta)$.

**2) MDP Constraints.** For the TB parametrization, where each trajectory is balanced, the MDP constraint loss is 0. For non-TB parametrization, we introduce a hyperparameter $\lambda_{\mathrm{OP}}$ to balance the order-preserving loss and MDP constraint loss. For the ease of discussion, we set $\lambda_{\mathrm{OP}} = 1$ for TB parametrization. We defer the detailed descriptions to Appendix C.

### 3.2 SINGLE-OBJECTIVE MAXIMIZATION

In the single-objective maximization, i.e. $D = 1$, there exists a global ordering, or *ranking* induced by $u(x)$, among all the candidates. We consider the scenario where the GFlowNet is used to sample $\arg\max_{x\in\mathcal{X}} u(x)$. We assume the local labeling is defined on pairs, i.e., $X = (x, x')$. Therefore, from Equation (1), the labeling and local distributions are,

$$\mathbb{P}_y(x|X) = \frac{\mathbf{1}(u(x) > u(x')) + \mathbf{1}(u(x) \geq u(x'))}{2}, \quad \mathbb{P}(x|X, \widehat{R}) = \frac{\widehat{R}(x)}{\widehat{R}(x) + \widehat{R}(x')},$$

and consequently we can calculate Equation (2) of $\mathcal{L}_{\mathrm{OP}}(X = (x, x'); \widehat{R})$.

In the following, we provide some theoretical analysis under the single-objective maximization task. We defer our experimental verifications to Figure E.6 in Appendix E.2.3, where we visualize the average of learned $\widehat{R}(\cdot)$ on states of the same objective values.

**1) Order-Preserving.** We claim that the learned reward $\widehat{R}(x_i)$ is a piece-wise geometric progression with respect to the ranking $i$. We first consider the special case: when $u(x_i)$ is mutually different, $\log \widehat{R}(x_i)$ is an arithmetic progression.

**Proposition 1** (Mutually different). *For $\{x_i\}_{i=0}^{n} \in \mathcal{X}$, assume that $u(x_i) < u(x_j), 0 \leq i < j \leq n$. The order-preserving reward $\widehat{R}(x) \in [1/\gamma, 1]$ is defined by the reward function that minimizes the order-preserving loss for neighboring pairs $\mathcal{L}_{\mathrm{OP-N}}$, i.e.,*

$$\widehat{R}(\cdot) := \arg\min_{r, r(x) \in [1/\gamma, 1]} \mathcal{L}_{\mathrm{OP-N}}(\{x_i\}_{i=0}^{n}; r) := \arg\min_{r, r(x) \in [1/\gamma, 1]} \sum_{i=1}^{n} \mathcal{L}_{\mathrm{OP}}(\{x_{i-1}, x_i\}; r). \tag{4}$$

*We have $\widehat{R}(x_i) = \gamma^{i/n-1}, 0 \leq i \leq n$, and $\mathcal{L}_{\mathrm{OP-N}}(\{x_i\}_{i=0}^{n}; \widehat{R}) = n \log(1 + 1/\gamma)$.*

In practice, $\gamma$ is not a fixed value. Instead, $\gamma$ is driven to infinity when minimizing $\mathcal{L}_{\mathrm{OP-N}}$ with a variable $\gamma$. Combined with Proposition 1, we claim that the order-preserving loss gradually sparsifies the learned reward $\widehat{R}(x)$ on mutually different objective $u(x)$ during training, where $\widehat{R}(x_i) \to 0, i \neq n$.

We also consider the more general setting, where $\exists i \neq j$, such that $u(x_i) = u(x_j)$. We first give an informal proposition in Proposition 2, and defer its formal version and proof to Proposition 5.

**Proposition 2** (Informal). *For $\{x_i\}_{i=0}^{n} \in \mathcal{X}$, assume that $u(x_i) \leq u(x_j), 0 \leq i < j \leq n$. Using the notations in Proposition 1, when $\gamma$ is sufficiently large, there exists $\alpha_\gamma$, $\beta_\gamma$, dependent on $\gamma$, such that $\widehat{R}(x_{i+1}) = \alpha_\gamma \widehat{R}(x_i)$ if $u(x_{i+1}) > u(x_i)$, and $\widehat{R}(x_{i+1}) = \beta_\gamma \widehat{R}(x_i)$ if $u(x_{i+1}) = u(x_i)$, for $0 \leq i \leq n - 1$. Also, minimize the $\mathcal{L}_{\mathrm{OP-N}}$ qith a variable $\gamma$ will drive $\gamma \to \infty, \alpha_\gamma \to \infty, \beta_\gamma \to 1$.*

If we do not fix a positive lower bound $1/\gamma$, i.e., let $\widehat{R}(x) \in [0, 1]$, minimizing $\mathcal{L}_{\mathrm{OP-N}}$ defined in Equation (4) will drive $\alpha_\gamma \to \infty, \beta_\gamma \to 1$ as $\gamma \to \infty$, which indicates $\widehat{R}(x_j)/\widehat{R}(x_i) \to 1$ if $u(x_i) = u(x_j)$, and $\widehat{R}(x_j)/\widehat{R}(x_i) \to \infty$ if $u(x_i) < u(x_j)$ as training goes on, i.e. $\widehat{R}(\cdot)$ enlarges the relative gap between different objective values, and assign similar reward to states of the same objective values. Since the GFlowNet sampler samples candidates with probabilities in proportion to $\widehat{R}(\cdot)$, it can sample high objective value candidates with a larger probability as the training progresses. We remark that the sparsification of the learned reward and the training of the GFlowNet happens simultaneously. Therefore, the exploration on the early training stages and the exploitation on the later sparsified reward $\widehat{R}(\cdot)$ can both be achieved.

**(B) MDP Constraints.** In this part, we match the flow $F(\cdot)$ with $\widehat{R}(\cdot)$, where $\widehat{R}(\cdot)$ is fixed and learned in Proposition 2. We consider the sequence prepend/append MDP (Shen et al., 2023).

**Definition 1** (Sequence prepend/append MDP). *In this MDP, states are strings, and actions are prepending or appending a symbol in an alphabet. This MDP generates strings of length $l$.*

In the following proposition, we claim that matching the flow $F(\cdot)$ with a sufficiently trained OP-GFN (i.e. sufficiently large $\gamma$) will assign high flow value on non-terminal states that on the trajectories ending in maximal reward candidates.

**Proposition 3.** *In the MDP in Definition 1, we consider the dataset $\{x_i, x'_n\}_{i=0}^n$ with $u(x_0) < u(x_1) < \cdots < u(x_n) = u(x'_n)$. We define the important substring $s^\star$ as the longest substring shared by $x_n, x'_n$, and $s_k(x)$ as the set of $k$-length substrings of $x$. Following Proposition 5, let the order-preserving reward $\widehat{R}(\cdot) \in [1/\gamma, 1]$, and the ratio $\alpha_\gamma$. We fix $P_B$ to be uniform, and match the flow $F(\cdot)$ with $\widehat{R}(\cdot)$ on terminal states. Then, when $\alpha_\gamma > 4$, we have $\mathbb{E}F(s^\star) > \mathbb{E}F(s), \forall s \in s_{|s^\star|}(x) \backslash s^\star$, where the expectation is taken over the random positions of $s^\star$ in $x_n, x'_n$.*

### 3.3 EVALUATION

#### 3.3.1 SINGLE-OBJECTIVE

GFlowNets for single-objective tasks are typically evaluated by measuring their ability to match the target distribution, e.g. by using the Spearman correlation between $\log p(x)$ and $R(x)$ (Madan et al., 2022; Nica et al., 2022). However, our method learns a GFlowNet to sample the terminal states proportional to the reward $\widehat{R}(x)$ instead of $R(x)$, which is unknown in prior. Therefore, we only focus on evaluating the GFlowNet's ability to discover the maximal objective.

#### 3.3.2 MULTI-OBJECTIVE

The evaluation of the multi-objective sampler is significantly more difficult. We assume the reference set $P := \{\boldsymbol{p}_j\}_{j=1}^{|P|}$ is the set of the true Pareto front points, and $S = \{\boldsymbol{s}_i\}_{i=1}^{|S|}$ is the set of all the generated candidates, $P' = \mathrm{Pareto}(S)$ is non-dominated points in $S$. When the true Pareto front is unknown, we use a discretization of the extreme faces of the objective space hypercube as $P$.

Audet et al. (2021) summarized various performance indicators to measure the convergence and uniformity of the generated Pareto front. Specifically, we measure: 1) the convergence of all the generated candidates $S$ to the true Pareto front $P$, by **Averaged Hausdorff distance**; 2) the convergence of the estimated Pareto front $P'$ to the true Pareto front $P$, by **IGD+, HyperVolume, $R_2$ indicator**. 3) the uniformity of the estimated front $P'$ with respect to $P$, by **PC-ent, $R_2$ indicator**. We discuss these metrics in Appendix F.1. During the computation on the estimated Pareto front, we will not de-duplicate the identical objective vectors.

## 4 SINGLE-OBJECTIVE EXPERIMENTS

### 4.1 HYPERGRID

We study a synthetic HyperGrid environment introduced by (Bengio et al., 2021a). In this environment, the states $\mathcal{S}$ form a $D$-dimensional HyperGrid with side length $H$: $\mathcal{S} = \{(s^1, \ldots, s^D) \mid (H-1) \cdot s^d \in \{0, 1, \ldots, H-1\}, d = 1, \ldots, D\}$, and non-stop actions are operations of incrementing one coordinate in a state by $\frac{1}{H-1}$ without exiting the grid. The initial state is $(0, \ldots, 0)$. For every state $s$, the terminal action is $s \to s^\top = x$. The objective at the state $x = (x^1, \ldots, x^D)^\top$ is given by $u(x) = R_0 + 0.5 \prod_{d=1}^D \mathbb{I}\left[\left|x^d - 0.5\right| \in (0.25, 0.5]\right] + 2\prod_{d=1}^D \mathbb{I}\left[\left|x^d - 0.5\right| \in (0.3, 0.4)\right]$, where $\mathbb{I}$ is an indicator function and $R_0$ is a constant controlling the difficulty of exploration. The ability of standard GFlowNets, i.e. $R(x) = u(x)$, is sensitive to $R_0$. Large $R_0$ facilitates exploration but hinders exploitation, whereas low $R_0$ facilitates exploitation but hinders exploration. However, OP-GFNs only deal with the pairwise order relation, so is independent of $R_0$.

We set $(D, H) = (2, 64), (3, 32)$ and $(4, 16)$, and compare TB and order-preserving TB (OP-TB). For $(D, H) = (2, 64)$, we plot the observed distribution on 4000 most recently visited states in Figure E.4. We consider the following three ratios to measure exploration-exploitation: 1) #(distinctly visited states)/#(all the states); 2) #(distinctly visited maximal states)/ #(all the maximal states); 3) In the most recently 4000 visited states, #(distinctly maximal states)/4000 in Figure E.5. A good

sampling algorithm should have a small ratio 1), and a large ratio 2), 3). We confirm that TB's performance is sensitive to the choice of $R_0$, and observe that OP-TB can recover all maximal areas more efficiently, and sample maximal candidates with higher probability after visiting fewer distinct candidates. The detailed discussions are in Appendix E.2.2. To conclude, OP-GFNs can balance exploration and exploitation without the selection of $R_0$.

## 4.2 MOLECULAR DESIGN

We study various molecular designs environments (Bengio et al., 2021a), including **Bag, TFBind8, TFBind10, QM9, sEH**, whose detailed descriptions are in Appendix E.3.1. We use the the sequence formulation (Shen et al., 2023) for the molecule graph generation, i.e., sequence prepend/append MDP in Definition 1. We define the optimal candidates in Bag as the $x$ with objective value 30, in SIX6, PHO4, QM9 as the top 0.5% $x$ ranked by the objective value, in sEH as the top 0.1% $x$ ranked by the objective value. The total number of such candidates for Bag, SIX6, PHO4, QM9, sEH is 3160, 328, 5764, 805, 34013 respectively.

We consider previous GFN methods and reward-maximization methods as baselines. Previous GFN methods include TB, DB, subTB, maximum entropy (MaxEnt, Malkin et al. (2022)), and substructure-guided trajectory balance (GTB, Shen et al. (2023)). For reward-maximization methods, we consider a widely-used sampling-based method in the molecule domain, Markov Molecular Sampling (MARS, Xie et al. (2021)), and RL-based methods, including actor-critic (A2C, Mnih et al. (2016)), Soft Q-Learning (SQL, Hou et al. (2020)), and proximal policy optimization (PPO, Schulman et al. (2017)). We run the experiments over 3 seeds, and plot the mean and variance of the objective value of top-100 candidates ranked by $u(x)$, and also plot the number of optimal candidates being found among all the generated candidates in Figure 4.1. We find that the order-preserving method outperforms all the baselines in both the ability to find the number of different optimal candidates and the average top-$k$ performance.

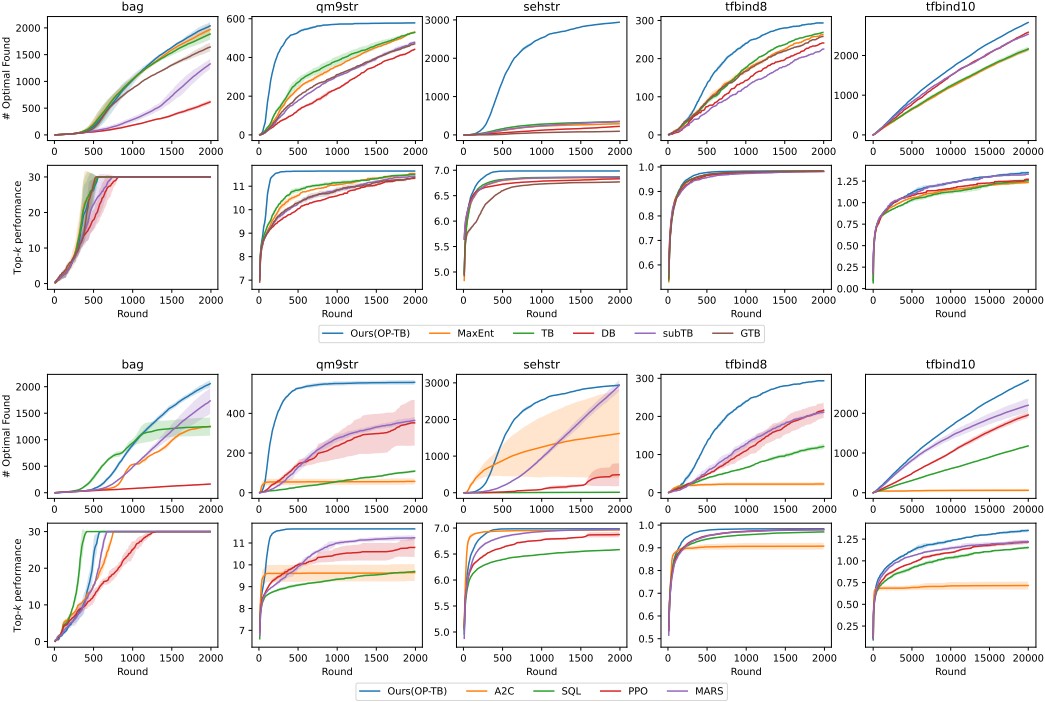

Figure 4.1: **Molecular design**: In the environment Bag, QM9, sEH, TFBind8, TFBind10, we test our algorithm (OP-TB) against previous GFN methods (MaxEnt, TB, DB, subTB, GTB), and (RL-)sampling methods (MARS, A2C, SQL, PPO).

### 4.3 NEURAL ARCHITECTURE SEARCH

#### 4.3.1 NAS ENVIRONMENT

We study the neural architecture search environment *NATS-Bench* (Dong et al., 2021), which includes three datasets: CIFAR10, CIFAR-100 and ImageNet-16-120. We choose the *topology search space* in NATS-Bench. The neural architecture search can be regarded as a sequence generation problem, where each sequence of neural network operations uniquely determines an architecture. In the MDP, each forward action fills any empty position in the sequence, starting from the empty sequence. For sequence $x \in \mathcal{X}$, the objective function $u_T(x)$ is the test accuracy of $x$'s corresponding architecture with the weights at the $T$-th epoch during its standard training pipeline. To measure the cost to compute $u_T(\cdot)$, we introduce the simulated train and test (T&T) time, which is defined by the time to train the architecture to the epoch $T$ and evaluate its test accuracy at epoch $T$. NATS-Bench provides APIs on $u_T(\cdot)$ and its T&T time for $T \leq 200$. Following Dong et al. (2021), when training the GFlowNet, we use the test accuracy at epoch 12 ($u_{12}(\cdot)$) as the objective function in training; when evaluating the candidates, we use the test accuracy at epoch 200 ($u_{200}(\cdot)$) as the objective function in testing. We remark that $u_{12}(\cdot)$ is a proxy for $u_{200}(\cdot)$ with lower T&T time, and the global ordering induced by $u_{12}(\cdot)$ is also a proxy for the global ordering induced by $u_{200}(\cdot)$. OP-GFNs only access the ordering of $u_{12}(\cdot)$, ignoring the unnecessary information of $u_{12}(\cdot)$'s exact value.

#### 4.3.2 EXPERIMENTAL DETAILS

We train the GFlowNet in a multi-trial sampling procedure described in Appendix E.4.2, and optionally use the backward KL regularization (-KL) and trajectories augmentation (-AUG) introduced in Appendix E.1. To monitor the training process, in each training round, we will record the architecture that has the highest objective function in training and the accumulated T&T time up to that round. We terminate the training when the accumulated T&T time reaches the threshold, of 50000, 100000 and 200000 seconds for CIFAR10, CIFAR100, and ImageNet-16-120 respectively. We adopt the RANDOM as baselines, and compare our results against previous multi-trial sampling methods: 1) evolutionary strategy, e.g., REA (Real et al., 2019); 2) reinforcement learning (RL)-based methods, e.g., REINFORCE (Williams, 1992), 3) HPO methods, e.g., BOHB (Falkner et al., 2018), whose experimental settings are described in Appendix E.4.2. To evaluate the algorithms, we plot the averaged accuracy (at epochs 12 and 200) of the recorded sequence of architectures with respect to the recorded sequence of accumulated T&T time, over 200 random seeds. [3] We report the results on trajectory balance (TB) and its order-preserving variants (OP-TB) in Figure 4.2[4], and defer the results of non-TB methods to Appendix E.4.5, where we include the ablation studies of (OP-)DB, (OP-)FM, (OP-)subTB, and the hyperparameters $\lambda_{\text{OP}}$. We observe that OP-non-TB methods can achieve similar performance gain with OP-TB, which validates the effectiveness and generality of order-preserving methods. Moreover, we compare the OP-TB with the GFN-$\beta$ algorithm in Appendix E.4.4, where $\beta = 4, 8, 16, 32, 64, 128$, and plot the results in Figure E.8. We observe that the OP-GFN outperforms the GFN-$\beta$ method by a significant margin.

We conclude that order-preserving GFlowNets consistently improve over the previous baselines in both the objective functions used in training and testing, especially in the early training stages. Besides, backward KL regularization and backward trajectory augmentation also contribute positively to the sampling efficiency. Finally, once we get a trained GFlowNet sampler, we can also use the learned order-preserving reward as a proxy to further boost the sampling efficiency, see Appendix E.4.3.

## 5 MULTI-OBJECTIVE EXPERIMENTS

### 5.1 HYPERGRID

We study HyperGrid environment in Section 4.1 with $(D, H) = (2, 32)$, and four normalized objectives: `brannin`, `currin`, `shubert`, `beale`, see Appendix F.2 for details. All the objectives are normalized to fall between 0 and 1. The true Pareto front of HyperGrid environment can be explicitly obtained by enumerating all the states.

---

[3]Since different runs sample different sequences of architectures, and different architectures have different T&T times, each run's sequence of time coordinates may not be uniformly spaced. We first linearly interpolate each run's sequences of points and then calculate the mean and variance on some fixed reference points.

[4]We remark that the first 64 candidates of GFN methods are generated by random policy. The point at which we observe a sudden performance increase of the GFN methods indicates the start of the training.

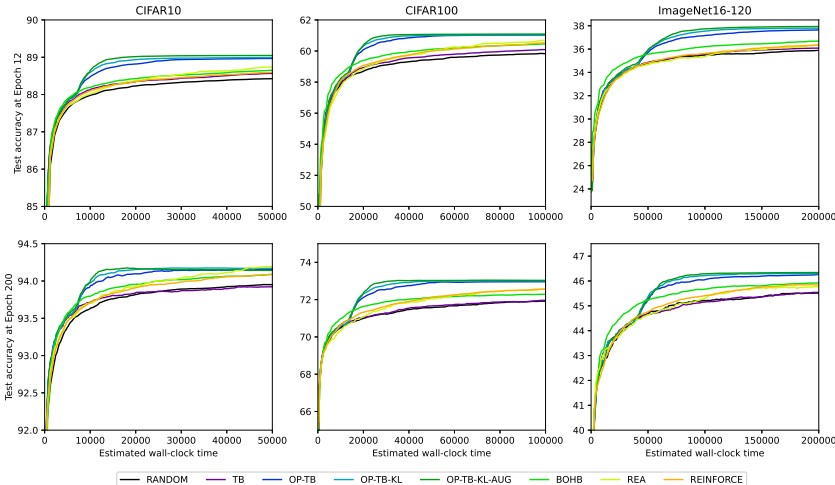

Figure 4.2: **Multi-trial training of a GFlowNet sampler**. Best test accuracy at epoch 12 and 200 of random baseline (Random), GFlowNet methods (TB, OP-TB, OP-TB-KL, OP-TB-KL-AUG), and other multi-trial algorithms (REA, BOHB, REINFORCE).

The training procedures are described in Appendix F.2. We generate 1280 candidates by the learned GFlowNet sampler in the evaluation as $S$, and report the metrics in Table F.1. To visualize the sampler, we plot all the objective vectors, and the true Pareto front, as well as the first 128 generated objective vectors, and their estimated Pareto front, in the objective space $[0, 1]^2$ and $[0, 1]^3$ in Figure F.1. We observe that our sampler achieves better approximation (i.e. almost zero IGD+ and smaller $d_H$), and uniformity (i.e. higher PC-ent) to the Pareto front, especially in the non-convex Pareto front, such as `currin-shubert`. We also plot the learned reward distributions of OP-GFNs and compare them with the indicator functions of the true Pareto front solutions in Figure 5.1, where we observe that OP-GFNs can learn a highly sparse reward function that concentrates on the true Pareto solutions, outperforming PC-GFNs.

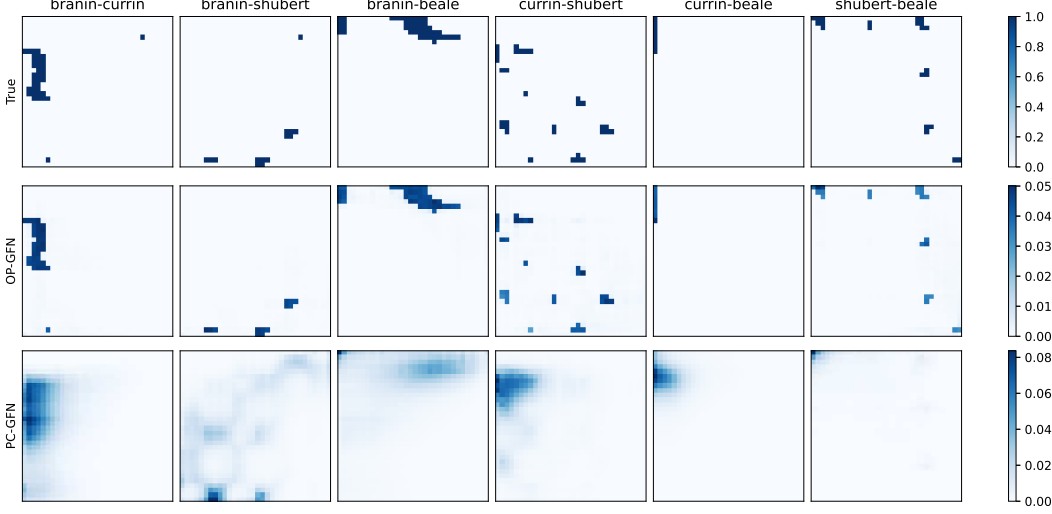

Figure 5.1: **Reward Distribution**: We plot the indicator function of the true Pareto front solutions and the learned reward distribution of the OP-GFNs and PC-GFNs.

## 5.2 N-GRAM

The synthetic sequence design task $n$-gram proposed by Stanton et al. (2022), is to generate sequences of a fixed maximum length $L = 36$. The vocabulary (action) to construct the sequence is of size 21, with 20 characters and a special token to end the sequence. The objectives are defined by the number

of occurrences of a given set of $n$-grams in a sequence $x$. We consider unigrams and bigrams in our experiments and summarize the objectives in Table F.2.

In the multi-objective optimization, we use a replay buffer to help stabilize the training (Roy et al., 2023, Appendix C.1). Specifically, instead of the on-policy update, we push the online sampled batch into the replay buffer, and immediately sample a batch of the same size to stabilize the training. We defer detailed experimental settings to Appendix F.3, and the experiment results are summarized in Table F.3. We observe that OP-GFNs achieve better performances on most of the tasks.

### 5.3 DNA SEQUENCE GENERATION

An instance illustrating a real-world scenario where the GFlowNet graph takes the form of a tree is the creation of DNA aptamers, which are single-stranded sequences of nucleotides widely employed in the realm of biological polymer design (Zhou et al., 2017; Yesselman et al., 2019). We generate the DNA sequences by adding one nucleobase ("A", "C", "T", "G") at a time, with a length of 30. We consider three objectives, (1) energy: the free energy of the secondary structure calculated by the software NUPACK (Zadeh et al., 2011); (2) pins: DNA hairpin index; (3) pairs: the number of base pairs. All the objectives are normalized to be bounded by 0 and 1. The experimental settings are detailed in Appendix F.4. We report the metrics in Table F.4, and plot the objective vectors in Figure F.2. We conclude that OP-GFNs achieve similar or better performance than preference conditioning, especially in the diversity of the estimated Pareto front.

### 5.4 FRAGMENT-BASED MOLECULE GENERATION

The fragment-based molecule generation is a four-objective molecular generation task, (1) qed: the well-known drug-likeness heuristic QED (Bickerton et al., 2012); (2) seh: the sEH binding energy prediction of a pre-trained publicly available model (Bengio et al., 2021a); (3) sa: a standard heuristic of synthetic accessibility; (4) mw: a weight target region penalty, which favors molecules with a weight of under 300. All the objectives are normalized to be bounded by 0 and 1.

We compare our OP-GFNs to both the preference (PC) and goal conditioning (GC) GFN. To stabilize the training of OP-GFNs and GC-GFNs, we use the same replay buffer as in Section 5.2. The detailed experimental settings are in Appendix F.5. In evaluation, we sample 64 candidates per round, 50 rounds using the trained sampler. We plot the estimated Pareto front in Figure 5.2, and defer the full results in Figure F.3 and Table F.5. We conclude that OP-GFNs achieve comparable or better performance with condition-based GFNs without scalarization in advance.

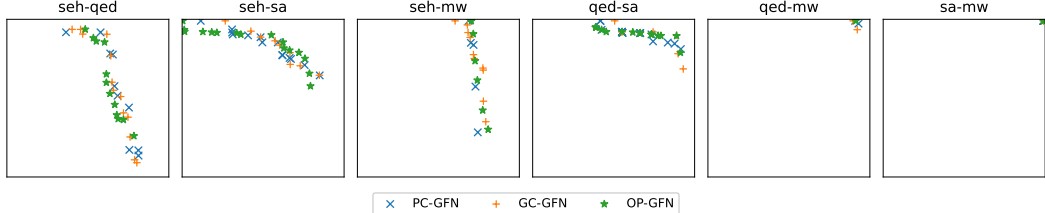

Figure 5.2: **Fragment-Based Molecule Generation**: We plot the estimated Pareto front of the generated samples in $[0, 1]^2$. The $x$-, $y$-axis are the first, and second objective in the title of respectively.

## 6 CONCLUSION

In this paper, we propose the order-preserving GFlowNets that sample composite objects with probabilities in proportion to a learned reward function that is consistent with a provided (partial) order. We theoretically prove that OP-GFNs learn to sample optimal candidates exponentially more often than non-optimal candidates. The main contribution is that our method does not require an explicit scalar reward function in prior, and can be directly used in the MOO tasks without scalarization. Also, when evaluating the objective function's value is costly, but the ordering relation, such as the pairwise comparison is feasible, OP-GFNs can efficiently reduce the training cost.

We will continue exploring the order-based sampling methods, especially in the MOO tasks. For example, we currently resample from the replay buffer to ensure that the training of OP-GFNs does not collapse to part of the Pareto front. In the future, we hope that we can introduce more controllable guidance to ensure the diversity of the OP-GFNs' sampling.

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

## A    OVERVIEW OF APPENDIX

We give a brief overview of the appendix here.

- **Appendix B**: we discuss related work on reinforcement learning, discrete GFlowNets and NAS Benchmark.
- **Appendix C**: we incorporate order-preserving loss with GFlowNets criteria other than trajectory balance.
- **Appendix D**: we provide the formal descriptions and missing proofs of Proposition 1, Proposition 2 and Proposition 3 in Section 3.2.
- **Appendix E** Complete single-objective experiments.
    - **Appendix E.1** Preliminaries and tricks in the single-objective experiments.
    - **Appendix E.2, HyperGrid Experiments**: In Appendix E.2.1, we validate the effectiveness of backward KL regularization proposed in Appendix E.1. In Appendix E.2.2, we provide the ablation study of $R_0$. In Appendix E.2.3, we plot the learned reward distribution.
    - **Appendix E.3, Molecule Experiments**: we provide the implementation details for each environment and present complete experimental results.
    - **Appendix E.4, NAS Experiments**: In Appendix E.4.1, we give a complete description of the NAS environment. In Appendix E.4.2, we provide the implementation details. In Appendix E.4.3, we provide experimental results on boosting the sampler. In Appendix E.4.4, we provide the ablation study on GFN-$\beta$ methods, KL regularization hyperparameter $\lambda_{\mathrm{KL}}$, and the size of the randomly generated dataset at initialization. In Appendix E.4.5, we provide the experimental results on OP methods in FM, DB and subTB.
- **Appendix F**: Complete multi-objective experiments.

    In **Appendix F.1**, we discuss multiple multi-objective evaluation metrics. In **Appendix F.2**, **Appendix F.3**, **Appendix F.4**, **Appendix F.5**, we conduct experiments on HyperGrid, n-gram, DNA sequence generation, and fragment-based molecule generation separately.

## B    RELATED WORK

**Reinforcement Learning** GFlowNets are trained to sample proportionally the reward rather than maximize it in standard RL. However, on tree-structured DAGs (autoregressive generation) are equivalent to RL with appropriate entropy regularization or soft Q-learning and control as inference (Buesing et al., 2020; Haarnoja et al., 2017; 2018). The experiments and theoretical explanation of Bengio et al. (2021a) show how standard RL-based methods can fail in the general DAG case, while GFlowNets can handle it well. Signal propagation over sequences of several actions in trajectory balance is also related to losses used in RL computed on subtrajectories (Nachum et al., 2017). However, compared with our proposed OP-GFNs, previous RL baselines are limited in their application to the multi-objective optimization and are inferior in the diversity of the solutions.

**Discrete GFlowNets** GFlowNets were first formulated as a reinforcement learning algorithm (Bengio et al., 2021a), with discrete state and action spaces, that trains a sequential sampler that samples the terminal states with probabilities in proportion to a given reward by the flow matching objective. Bengio et al. (2021b) provides the theoretical foundations of GFlowNets, based on flow networks defined on MDPs, and proposes the detailed balance loss that bypasses the need to sum the flows over large sets of children and parents. Later, trajectory balance (Malkin et al., 2022), sub-trajectory balance (Madan et al., 2022), augmented flow (Pan et al., 2022), forward-looking (Pan et al., 2023), quantile matching (Zhang et al., 2023c), local search (Kim et al., 2023b) were proposed to improve the credit assignments along trajectories.

**NAS Benchmark** The first tabular NAS benchmark to be released was NAS-Bench-101 (Ying et al., 2019). This benchmark consists of 423,624 architectures trained on CIFAR-10. NAS-Bench-201 (Dong & Yang, 2020) is another popular tabular NAS benchmark. The cell-based search space consists of a DAG where each edge can take on operations. The number of non-isomorphic architectures is 6,466 and all are trained on CIFAR-10, CIFAR-100, and ImageNet-16-120. NATS-Bench (Dong et al., 2021) is an extension of NAS-Bench-201, and provides an algorithm-agnostic

benchmark for most up-to-date NAS algorithms. The search spaces in NATS-Bench includes both architecture topology and size. The DARTS (Liu et al., 2018) search space with CIFAR-10, consisting of 1018 architectures, but is not queryable. 60 000 of the architectures were trained and used to create NAS-Bench-301 (Siems et al., 2020), the first surrogate NAS benchmark.

**Exploration in GFlowNets** . The exploration and exploitation strategy in GFlowNets has been analyzed recently, and we will discuss the difference here. . Rector-Brooks et al. (2023) demonstrates how Thompson sampling with GFlowNets allows for improved exploration and optimization efficiency in GFlowNets. We point out the key difference in the exploration strategy is that, Rector-Brooks et al. (2023) encourages the exploration by bootstrapping $K$ different policies, while in the early stages of OP-GFNs, the learned reward is almost uniform, which naturally facilitates the exploration. As the training goes on, the learned reward gets sparser, and the exploration declines and exploitation arises. We remark that the idea of Thompson sampling can be used in the exploitation stages of OP-GFNs to further encourage the exploration in the latter stages of training.

**Temperature-Conditional GFlowNets** . Zhang et al. (2023a); Kim et al. (2023a) propose the Temperature-Conditional GFlowNets (TC-GFNs) to learn the generative distribution with any given temperature. Zhang et al. (2023a) conditions the policy networks on the temperature, and Kim et al. (2023a) directly scales probability logits regarding the temperature. We point out some critical differences to OP-GFN: 1) In TC-GFNs, a suited $\beta$'s prior must still be chosen. while OP-GFNs do not require such a choice. 2) TC-GFNs learn to match $R^\beta(x)$ for all $\beta$, while OP-GFNs just learn to sample with the correct ordering statistics. However, using these few statistics, OP-GFNs still achieve competitive results in both single and multi-objective optimization. 3) TC-GFNs require the scalar reward function, while OP-GFNs can be directly used in multi-objective optimization.

## C   ORDER-PRESERVING AS A PLUGIN SOLVER

In the next, we discuss how we can integrate the loss $\mathcal{L}_{\mathrm{OP}}$ into existing (non-TB) training criteria. In this case, we will introduce a hyperparameter $\lambda_{\mathrm{OP}}$ to balance the order-preserving loss and the MDP constraint loss. Although $\lambda_{\mathrm{OP}}$ is dependent on the MDP structure and the objective function, we argue that OP-GFNs are less sensitive to different choices of $\lambda_{\mathrm{OP}}$ compared to the influence of different choices of $\beta$ on GFN-$\beta$ methods in Appendix E.4.5.

**Flow Matching.** In the parametrization of the flow matching, we denote the order-preserving reward by $\widehat{R}_{\mathrm{FM}}(x;\theta) = \sum_{(s''\to x)\in\mathcal{A}} F(s'',x;\theta)$. The loss can be written as:

$$\mathcal{L}_{\mathrm{OP-FM}}(T_B;\theta) = \mathbb{E}_{\tau_i\sim T_B} \sum_{t=1}^{n_i-1} \mathcal{L}_{\mathrm{FM}}(s_t^i;\theta) + \lambda_{\mathrm{OP}} \cdot \mathcal{L}_{\mathrm{OP}}(X_B;\widehat{R}_{\mathrm{FM}}(\cdot;\theta)),$$

where

$$\mathcal{L}_{\mathrm{FM}}(s;\theta) = \left(\log \sum_{(s''\to s)\in\mathcal{A}} F(s'',s;\theta) - \log \sum_{(s\to s')\in\mathcal{A}} F(s,s';\theta)\right)^2,$$

is the flow matching loss for each state $s$.

**Detailed Balance.** In the parametrization of the detailed balance, we denote the order-preserving reward at terminal states by $\widehat{R}_{\mathrm{DB}}(x;\theta) = F(x;\theta), x \in \mathcal{X}$. The loss can be written as:

$$\mathcal{L}_{\mathrm{OP-DB}}(T_B;\theta) = \mathbb{E}_{\tau_i\sim T_B} \sum_{t=1}^{n_i-1} \mathcal{L}_{\mathrm{DB}}(s_{t-1}^i, s_t^i;\theta) + \lambda_{\mathrm{OP}} \cdot \mathcal{L}_{\mathrm{OP}}(X_B;\widehat{R}_{\mathrm{DB}}(\cdot;\theta)),$$

where

$$\mathcal{L}_{\mathrm{DB}}(s,s';\theta) = (\log F(s;\theta)P_F(s'|s;\theta) - \log F(s';\theta)P_B(s|s';\theta))^2,$$

is the detailed balance loss for each transition $s \to s'$.

**SubTrajectory Balance.** Similar to the detailed balance, the order-preserving reward is also $\widehat{R}_{\mathrm{subTB}}(x;\theta) = F(x;\theta), x \in \mathcal{X}$. The loss can be written as

$$\mathcal{L}_{\mathrm{OP-subTB}}(T_B;\theta) = \mathbb{E}_{\tau_i\sim T_B} \sum_{0\le u<v\le n_i} \lambda_{\mathrm{subTB}}^{u-v}\mathcal{L}_{\mathrm{subTB}}(\tau_{u,v}^i;\theta) + \lambda_{\mathrm{OP}} \cdot \mathcal{L}_{\mathrm{OP}}(X_B;\widehat{R}_{\mathrm{subTB}}(\cdot;\theta)),$$

where $\lambda_{\mathrm{subTB}}$ is the subtrajectory geometric reweighting hyperparameter, and

$$\mathcal{L}_{\mathrm{subTB}}(\tau_{u,v};\theta) = \left( \log \frac{F(s_u;\theta) \prod_{t=u+1}^{v} P_F(s_t|s_{t-1};\theta)}{F(s_v;\theta) \prod_{t=u+1}^{v} P_B(s_{t-1}|s_t;\theta)} \right)^2,$$

is the subtrajectory balance loss for each subtrajectory $\tau_{u,v} := (s_u \to \cdots \to s_v), 0 \le u < v \le n$.

## D  MISSING PROOFS

For completeness, we also restate Proposition 1 and Proposition 3 here in Proposition 4 and Proposition 6 respectively.

**Proposition 4** (Mutually different)**.** *For $\{x_i\}_{i=0}^n \in \mathcal{X}$, assume the objective function $u(x)$ is known, and $u(x_i) < u(x_j), 0 \le i < j \le n$. The order-preserving reward $\widehat{R}(x) \in [1/\gamma, 1], 0 < a < b$, is defined by the reward function that minimizes the order-preserving loss for neighboring pairs $\mathcal{L}_{\mathrm{OP-N}}$:*

$$\widehat{R}(\cdot) := \arg\min_{r,r(x)\in[1/\gamma,1]} \mathcal{L}_{\mathrm{OP-N}}(\{x_i\}_{i=0}^n; r) := \arg\min_{r,r(x)\in[1/\gamma,1]} \sum_{i=1}^n \mathcal{L}_{\mathrm{OP}}(x_{i-1}, x_i; r). \quad (5)$$

*We have $\widehat{R}(x_i) = \gamma^{i/n-1}, 0 \le i \le n$, and $\mathcal{L}_{\mathrm{OP-N}}(\{x_i\}_{i=0}^n; \widehat{R}) = n\log(1+1/\gamma)$.*

*Proof of Proposition 4.* Let $\mathcal{L}_{\mathrm{OP}}(r) := \mathcal{L}_{\mathrm{OP-N}}(\{x_i\}_{i=0}^n, r)$ for abbreviation. Since the objective $\mathcal{L}_{\mathrm{OP}}(r)$ decreases as $r(x_0)$ decreases or $r(x_n)$ increases, we have $\widehat{R}(x_0) = 1/\gamma, \widehat{R}(x_n) = 1$. We denote $r(x_i)$ by $r_i, 0 \le i \le n$.

Let us consider $\widehat{R}(x_i), 1 \le i \le n-1$, since

$$\frac{\partial \mathcal{L}_{\mathrm{OP}}(r)}{\partial r_i} = \frac{1}{r_{i-1} + r_i} + \frac{1}{r_{i+1} + r_i} - \frac{1}{r_i} = 0 \iff r_i = \sqrt{r_{i-1} r_{i+1}}.$$

Therefore, $\widehat{R}(x_i) = \gamma^{\frac{i}{n}-1}, \quad 1 \le i \le n-1$, and the second order derivative

$$\frac{\partial^2 \mathcal{L}_{\mathrm{OP}}(r)}{\partial r_i^2}\Big|_{r_i=(\gamma)^{\frac{i}{n}-1}} = \frac{1}{r_i^2} \frac{2}{(1+1/\gamma)^2} > 0,$$

which proves it minimize $\mathcal{L}_{\mathrm{OP}}(r)$. $\square$

**Proposition 5.** *For $\{x_i\}_{i=0}^n \in \mathcal{X}$, assume the objective function $u(x)$ is known, and $u(x_i) \le u(x_j), 0 \le i < j \le n$. We define two subscript sets:*

$$I_1 := \{i : u(x_i) < u(x_i), 0 \le i \le n-1\}, \quad I_2 := \{i : u(x_i) = u(x_{i+1}), 0 \le i \le n-1\},$$

*We introduce two auxiliary states $x_{-1}$ and $x_{n+1}$ for boundary conditions, such that $u(x_{-1}) = -\infty, u(x_{n+1}) = +\infty$.[5] The order-preserving reward $\widehat{R}(x) \in [1/\gamma, 1]$ is defined by minimizing the order-preserving loss with neighboring states:*

$$\arg\min_{r,r(x)\in[1/\gamma,1]} \mathcal{L}_{\mathrm{OP-N}}(\{x_i\}_{i=0}^n \cup \{x_{-1}, x_{n+1}\}; r) := \arg\min_{r,r(x)\in[1/\gamma,1]} \sum_{i=-1}^{n+1} \mathcal{L}_{\mathrm{OP}}(x_{i-1}, x_i; r).$$

*Let $m = |I_1|$, and define one auxiliary function:*

$$f_1(\alpha) := \alpha^{m+2} \left(1 - \frac{4}{\alpha+3}\right)^{n-m}.$$

*Since $f_1(\alpha), \frac{1}{\gamma} f_1(\gamma^{\frac{1}{m+1}})$ are both monotonically increasing from 0 to infinity for $\alpha, \gamma \ge 1$. There exists unique $\gamma_0, \alpha_\gamma > 1$ such that*

$$f_1(\alpha_\gamma) = \gamma, \quad f_1(\gamma_0^{\frac{1}{m+1}}) = \gamma_0$$

*For $\gamma > \gamma_0$, we have*

$$\widehat{R}(x_0) = \alpha_\gamma \gamma^{-1}, \widehat{R}(x_{i+1}) = \alpha_\gamma \widehat{R}(x_i), i \in I_1, \quad \widehat{R}(x_{i+1}) = \beta_\gamma \widehat{R}(x_i), i \in I_2 \quad \beta_\gamma = \frac{\alpha_\gamma - 1}{\alpha_\gamma + 3}.$$

*Also, minimizing $\mathcal{L}_{\mathrm{OP-N}}$ will drive $\gamma \to +\infty$, and hence $\alpha_\gamma \to +\infty, \beta_\gamma \to 1$.*

---

[5]Note that for regular $x \in \mathcal{X}$, we have $u(x) \in [0, +\infty)$

*Proof of Proposition 5.* We can expand the order-preserving loss by separating two cases where $u(x_{i-1}) < u(x_i)$ or $u(x_{i-1}) = u(x_i)$ for $0 \le i \le n+1$. We remark that $u(x_{-1}) < u(x_0)$ and $u(x_n) < u(x_{n+1})$ by the definition of $x_{-1}$ and $x_{n+1}$.

$$\arg \min_{r, r(x) \in [1/\gamma, 1]} \mathcal{L}_{\text{OP-N}}(\{x_i\}_{i=0}^n \cup \{x_{-1}, x_{n+1}\}; r)$$

$$:= \arg \min_{r, r(x) \in [1/\gamma, 1]} \sum_{i=-1}^{n+1} \mathcal{L}_{\text{OP}}(x_{i-1}, x_i; r)$$

$$:= \arg \min_{r, r(x) \in [1/\gamma, 1]}$$

$$- \sum_{i \in I_1 \cup \{-1, n\}} \log \frac{r(x_{i+1})}{r(x_i) + r(x_{i+1})} - \frac{1}{2} \sum_{i \in I_2} \left( \log \frac{r(x_i)}{r(x_i) + r(x_{i+1})} + \frac{r(x_{i+1})}{r(x_i) + r(x_{i+1})} \right).$$

Let $\mathcal{L}_{\text{OP}}(r) := \mathcal{L}_{\text{OP-N}}(\{x_i\}_{i=0}^n \cup \{x_{-1}, x_{n+1}\}, r)$ for abbreviation. We denote $r(x_i)$ by $r_i$, $-1 \le i \le n+1$, and define $\alpha_{i-1}$ such that $r_i = \alpha_{i-1} r_{i-1}$, $0 \le i \le n+1$. Since the objective $\mathcal{L}_{\text{OP}}(r)$ decreases as $r_{-1}$ decreases or $r_{n+1}$ increases, we have $\widehat{R}(x_{-1}) = 1/\gamma$, $\widehat{R}(x_{n+1}) = 1$.

We consider the terms involving $r_i$, $0 \le i \le n$ in the order-preserving loss.

(1) If $u(x_{i-1}) = u(x_i) = u(x_{i+1})$, the relevant term is

$$\mathcal{L}_{\text{OP}}(r) = -\frac{1}{2} \left( \log \frac{r_{i-1}}{r_{i-1} + r_i} + \log \frac{r_i}{r_{i-1} + r_i} + \log \frac{r_{i+1}}{r_{i+1} + r_i} + \log \frac{r_i}{r_{i+1} + r_i} \right) + \cdots.$$

Then,

$$\frac{\partial \mathcal{L}_{\text{OP}}(r)}{\partial r_i} = \frac{1}{r_i + r_{i+1}} + \frac{1}{r_i + r_{i-1}} - \frac{1}{r_i}.$$

Setting $\frac{\partial \mathcal{L}_{\text{OP}}(r)}{\partial r_i} = 0$, we have

$$\alpha_i = \alpha_{i-1}.$$

(2) If $u(x_{i-1}) = u(x_i) < u(x_{i+1})$, the relevant term is

$$\mathcal{L}_{\text{OP}}(r) = -\frac{1}{2} \left( \log \frac{r_{i-1}}{r_{i-1} + r_i} + \log \frac{r_i}{r_{i-1} + r_i} \right) - \log \frac{r_{i+1}}{r_{i+1} + r_i} + \cdots.$$

Then,

$$\frac{\partial \mathcal{L}_{\text{OP}}(r)}{\partial r_i} = \frac{1}{r_i + r_{i+1}} - \frac{1}{2r_i} + \frac{1}{r_i + r_{i+1}}.$$

Setting $\frac{\partial \mathcal{L}_{\text{OP}}(r)}{\partial r_i} = 0$, we have

$$\alpha_i = 2 \cdot \frac{1 + \alpha_{i-1}}{1 - \alpha_{i-1}} - 1.$$

(3) If $u(x_{i-1}) < u(x_i) < u(x_{i+1})$, the relevant term is

$$\mathcal{L}_{\text{OP}}(r) = - \log \frac{r_i}{r_{i-1} + r_i} - \log \frac{r_{i+1}}{r_{i+1} + r_i} + \cdots.$$

Then,

$$\frac{\partial \mathcal{L}_{\text{OP}}(r)}{\partial r_i} = \frac{1}{r_i + r_{i+1}} + \frac{1}{r_i + r_{i+1}} - \frac{1}{r_i}.$$

Setting $\frac{\partial \mathcal{L}_{\text{OP}}(r)}{\partial r_i} = 0$, we have

$$\alpha_i = \alpha_{i-1}.$$

(4) If $u(x_{i-1}) < u(x_i) = u(x_{i+1})$, the relevant term is

$$\mathcal{L}_{\text{OP}}(r) = - \log \frac{r_i}{r_{i-1} + r_i} - \frac{1}{2} \left( \log \frac{r_i}{r_{i+1} + r_i} + \log \frac{r_{i+1}}{r_{i+1} + r_i} \right) + \cdots$$

Then,

$$\frac{\partial \mathcal{L}_{\mathrm{OP}}(r)}{\partial r_i} = \frac{1}{r_i + r_{i+1}} + \frac{1}{r_i + r_{i+1}} - \frac{3}{2r_i}.$$

Setting $\frac{\partial \mathcal{L}_{\mathrm{OP}}(r)}{\partial r_i} = 0$, we have

$$\alpha_i = \frac{\alpha_{i-1} - 1}{\alpha_{i-1} + 3}.$$

From (1), (2), (3), (4), assuming $u(x_{i-1}) < u(x_i) = \cdots = u(x_j) < u(x_{j+1})$, then

$$\alpha_j = 2 \cdot \frac{1 + \alpha_{j-1}}{1 - \alpha_{j-1}} - 1 = 2 \cdot \frac{1 + \alpha_i}{1 - \alpha_i} - 1 = 2 \cdot \frac{1 + \frac{\alpha_{i-1} - 1}{\alpha_{i-1} + 3}}{1 - \frac{\alpha_{i-1} - 1}{\alpha_{i-1} + 3}} - 1 = \alpha_{i-1}.$$

Therefore, we can define $\alpha := \alpha_i$ if $u(x_i) < u(x_{i+1})$, and $\beta := \alpha_i$ if $u(x_i) = u(x_{i+1})$, and $\beta = \frac{\alpha-1}{\alpha+3}, \alpha \geq 1, 0 \leq \beta \leq 1$. Then, $\log r$ is piecewise linear with two different slope, $\log \alpha$ and $\log \beta$. By the definition of $x_{-1}$ and $x_{n+1}$, we have $r_0 = \alpha r_{-1} = \alpha \gamma^{-1}, r_n = r_{n+1}/\alpha = b/\alpha = 1/\alpha$. According to the previous definition, We have $\alpha^{|I_1|+2} \beta^{|I_2|} = \gamma$, i.e. $\alpha$ is the solution to the following equation

$$\alpha^{m+2} \left( \frac{\alpha - 1}{\alpha + 3} \right)^{n-m} = \gamma.$$

Therefore, $\alpha = \alpha_\gamma$ by the definition of $\alpha_\gamma$. We finally need to ensure $\widehat{R}(\cdot)$ preserves the order, i.e. if $u(x_i) < u(x_j)$, we have $\widehat{R}(x_i) < \widehat{R}(x_j)$. We can bound:

$$\frac{\widehat{R}(x_j)}{\widehat{R}(x_i)} \geq \alpha_\gamma \left( \frac{\alpha_\gamma - 1}{\alpha_\gamma + 3} \right)^{n-m}. \tag{6}$$

and the equality holds iff $I_2 \subset \{i, i+1, \cdots, j-1\}$ and $j - i = |I_2| + 1$. We have by the definition of $\alpha_\gamma, \gamma_0$ and monotonic increasing of $f_1(\alpha)$ and $f_1(\gamma^{\frac{1}{m+1}})\gamma^{-1}$ w.r.t. $\alpha$ and $\gamma$,

$$\alpha_\gamma \left( \frac{\alpha_\gamma - 1}{\alpha_\gamma + 3} \right)^{n-m} > 1 \Longleftrightarrow \alpha_\gamma^{m+1} > \gamma \Longleftrightarrow f_1(\gamma^{\frac{1}{m+1}}) \geq \gamma \Longleftrightarrow \gamma > \gamma_0,$$

which satisfies the assumption, hence $\widehat{R}(x_j) > \widehat{R}(x_i)$.
The order-preserving loss can be explicitly written as:

$$\mathcal{L}_{\mathrm{OP}}(r) = -\frac{n-m}{2} \log \frac{\beta_\gamma}{(1+\beta_\gamma)^2} - (m+2) \log \frac{\alpha_\gamma}{1+\alpha_\gamma}$$

$$= -\frac{n-m}{2} \log \frac{(\alpha_\gamma - 1)(\alpha_\gamma + 3)}{4(\alpha_\gamma + 1)^2} - (m+2) \log \frac{\alpha_\gamma}{1+\alpha_\gamma},$$

and taking the derivative w.r.t. $\alpha_\gamma$, we have

$$\frac{\partial \mathcal{L}_{\mathrm{OP}}(r)}{\partial \alpha_\gamma} = -\frac{4n-m}{(\alpha_\gamma^2 - 1)(\alpha_\gamma + 3)} - \frac{m+2}{\alpha_\gamma(\alpha_\gamma + 2)} < 0.$$

Therefore, minimizing the order-preserving loss corresponds to making $\alpha_\gamma \to +\infty$, and therefore $\beta_\gamma \to 1$ and $\gamma \to +\infty$.

$\square$

**Remark.** *If $u(x_0) < u(x_1)$ or $u(x_{n-1}) < u(x_n)$, the auxiliary states $x_{-1}$ or $x_{n+1}$ are not necessary to be added. In this case, we have $u(x_0) = 1/\gamma$ or $u(x_0) = 1$ without auxiliary states, following a similar argument in Proposition 1. However, if there are multiple states of minimal or maximal objective value, we need to introduce $u(x_{-1}) = -\infty$ or $u(x_{n+1}) = +\infty$, so that $r_0$ or $r_n$ appears in*

*two terms in $\mathcal{L}_{\mathrm{OP}}(r)$, unifying our analysis and avoiding boundary condition difference. For example, if $u(x_0) = u(x_1)$ without $x_{-1}$, we have*

$$\frac{\partial \mathcal{L}_{\mathrm{OP}}(r)}{\partial r_0} = \frac{1}{r_0 + r_1} - \frac{1}{2r_0} + \frac{1}{r_0 + r_1},$$

$$\frac{\partial \mathcal{L}_{\mathrm{OP}}(r)}{\partial r_1} = \frac{1}{r_0 + r_1} - \frac{1}{2r_1} + \frac{1}{r_0 + r_1} + \frac{\mathcal{L}_{\mathrm{OP}}(x_1, x_2; r)}{\partial r_1},$$

*and $\frac{\partial \mathcal{L}_{\mathrm{OP}}(r)}{\partial r_0}$ and $\frac{\partial \mathcal{L}_{\mathrm{OP}}(r)}{\partial r_1}$ cannot be zero at the same time.*

**Proposition 6.** *In the sequence prepend/append MDP in Definition 1, we consider a fixed dataset $\{x_i, x'_i\}_{i=0}^n$ with $u(x_0) < u(x_1) < \cdots < u(x_n) = u(x'_n)$. Denote $s^\star$ as the important substring, defined as the longest substring shared by $x_n, x'_n$ with length $k$, and $s_k(x)$ as the set of $k$-length substrings of $x$. Following Proposition 5, let the order-preserving reward $\widehat{R} \in [1/\gamma, 1]$, and the ratio $\alpha_\gamma$. We fix $P_B$ to be uniform and match the flow $F(\cdot)$ with $\widehat{R}(\cdot)$ on terminal states. Then, when $\alpha_\gamma > 4$, we have $\mathbb{E}F(s^\star) > \mathbb{E}F(s), \forall s \in s_k(x) \backslash s^\star$, where the expectation is taken over the random positions of $s^\star$ in $x_n, x'_n$.*

*Proof of Proposition 6.* By Proposition 5, the reward $\widehat{R}(x_i) = \widehat{R}(x_0)\alpha_\gamma^i, \widehat{R}(x'_n) = \widehat{R}(x_i)\alpha_\gamma^n \beta_\gamma, 0 \leq i \leq n$. We claim that a uniform backward policy induces a uniform trajectory distribution over trajectories connecting $x$ to $s_0$. On constructing $x$, we have $n-1$ choices of prepending or appending. Therefore, there are $2^{l-1}$ trajectories ending at $x$, and each trajectory has flow $\frac{\widehat{R}(x)}{2^{l-1}}$.

Let $s$ of length $k$ is preceded by $a$ characters in $x$ of length $l$, in total there are $\binom{a}{l-k}2^{k-1}$ trajectories passing through $s$ that end at $x$. The average number of trajectories over a uniform distribution on $0 \leq a \leq l - k$ is $\frac{2^{l-1}}{l-k+1}$. Thus, the expected flow passing through $s$ and ending at just $x$, over uniformly random positions of $s$ in $x$ is $\frac{\widehat{R}(x)}{l-k+1}$ for any $s, x$.

We have $\mathbb{E}F(s^\star) \geq \frac{\widehat{R}(x_n) + \widehat{R}(x'_n)}{l-k+1}$, and for $s \in s_k(x_n) \backslash s^\star$, $\mathbb{E}F(s) \leq \frac{\widehat{R}(x_n) + \sum_{i=1}^{n-1} \widehat{R}(x'_i)}{l-k+1}$. As long as

$$1 + \alpha_\gamma + \cdots + \alpha_\gamma^{n-1} < \alpha_\gamma^n \beta_\gamma = \alpha_\gamma^n \left(1 - \frac{4}{\alpha_\gamma + 3}\right) \Longleftarrow \frac{1}{\alpha_\gamma - 1} < \frac{\alpha_\gamma - 1}{\alpha_\gamma + 3}.$$

i.e. $\alpha_\gamma > 4$ is sufficient to make the inequality hold.

we have $\mathbb{E}F(s^\star) > \mathbb{E}F(s), s \in s_k(x_n) \cup s_k(x'_n) \backslash s^\star$. Therefore, then the order-preserving reward can help correctly assign credits to the high-reward intermediate state $s'$. Note our results does not contradict with those in Shen et al. (2023), since we are considering $F(s), s \in s_k(x) \backslash s^\star$, instead of $F(s_k(x) \backslash s^\star)$. $\qquad\square$

# E  SINGLE-OBJECTIVE EXPERIMENTS

In this section, our implementation is based on torchgfn (Lahlou et al., 2023), Shen et al. (2023)'s implementation.

## E.1  PRELIMINARIES

We introduce some important tricks we used to improve the performance in the experiments.

**Backward KL Regularization** Fixed uniform backward distribution $P_B$ has been shown to avoid bias induced by joint training of $P_B$ and $P_F$, but also suffers from the slow convergence (Malkin et al., 2022). In this paper, we propose the regularize the backward distribution by its KL divergence w.r.t. uniform distribution. For a trajectory $\tau = (s_0 \to s_1 \to \cdots \to s_n = x)$, define the *KL regularized trajectory loss $\mathcal{L}_{\mathrm{KL}}$* as

$$\mathcal{L}_{\mathrm{KL}}(\tau) := \frac{1}{n} \sum_{i=1}^n \mathcal{L}_{\mathrm{KL}}(s_t), \quad \text{where } \mathcal{L}_{\mathrm{KL}}(s_t) := \mathrm{KL}(P_B(\cdot|s_t; \theta) \| U_B(\cdot|s_t)),$$

where $U_B(\cdot|s_t)$ is uniform distribution on valid backward actions. Such KL regularizer can be plugged into any training objective that parametrizes the backward probability $P_B$, which includes DB and (sub)TB objective. In Appendix E.2.1, we show that KL regularizer provides the advantages of both fixed and trainable $P_B$.

**Backward Trajectories Augmentation** We adopt the prioritized replay training (PRT) (Shen et al., 2023), that focuses on high reward data. We form a replay batch from $\mathcal{X}$, all terminal states seen so far, so that $\alpha_1$ percentile of the batch is sampled from the top $\alpha_2$ percentile of the objective function, and the rest of the batch is sampled from the bottom $1 - \alpha_2$ percentile. We sample augmented trajectories from the replay batch using $P_B$. In practice, we select $\alpha_1 = 50, \alpha_2 = 10$.

$\widehat{R}(x)$ **as a Proxy** When evaluating the objective function $u(x)$ is costly, we propose to use $\widehat{R}(x)$ as a proxy. If we want to sample $k$ terminal states with maximal rewards, we can first sample $K \gg k$ terminal states, and pick states with top-$k$ $\widehat{R}(x)$. Then, we need only evaluate $u(x)$ on $k$ instead of $K$ terminal states. We define the *ratio* of boosting to be $r_{\text{boost}} = K/k$. For GFlowNet objective parametrize $F(s)$, we can directly let $\widehat{R}(x) = F(x), x \in \mathcal{X}$. For TB objective, we need to use Equation (3) to approximate $\widehat{R}(x)$. Since the cost of evaluating $\widehat{R}(x)$ is also non-negligible, we only adopt this strategy when obtaining $u(x)$ directly is significantly more difficult. For example, in the neural architecture search environment (in Section 4.3), evaluating $u_T(x)$ requires training a network to step $T$ to get the test accuracy.

**Training Procedure** We adopt the hybrid of online and offline training of the GFlowNet. The full pseudo algorithm is summarized in Algorithm 1.

---

**Algorithm 1** Order-Preserving GFlowNet

    **Inputs:**
        $N_{\text{init}}$: number of forward sampled terminal states at initialization.
        $N_{\text{round}}$: number of rounds for forward sampling.
        $N_{\text{new}}$: number of forward sampled terminal states in every round.
        $N_{\text{off}}$: number of backward augmented terminal states.
        $N_{\text{off}-\text{per}}$: number of backward augmented trajectories per terminal state.
    **Initialize:**
        $\mathcal{T}_0$: Random initialized trajectories of size $N_{\text{init}}$.
        $\theta_0$: GFlowNet with parameters $\theta = \theta_0$:
        $L(\theta; \mathcal{T})$: Order preserving loss defined in Equation (2).
    **for** $i = 1 \rightarrow N_{\text{round}}$ **do**
        Update parameters $\theta_i' \leftarrow \theta_{i-1}$ on trajectories set $\mathcal{T}_{i-1}$.
        Sample $N_{\text{new}}$ terminal trajectories $\mathcal{T}_i'$ with the forward action sampler parameterized by $\theta_i'$.
        Update trajectory sets $\mathcal{T}_i \leftarrow \mathcal{T}_{i-1} \cup \mathcal{T}_i'$.
        Sample $N_{\text{off}}$ terminal states from $\mathcal{T}_i$. Augment into trajectories $\mathcal{T}_i''$ by the uniform backward sampler, with $N_{\text{off}-\text{per}}$ trajectories per terminal state.
        Update parameters $\theta_i \leftarrow \theta_i'$ on trajectories set $\mathcal{T}_i''$.
    **end for**

---

## E.2 HYPERGRID

**Objective Function** The objective function at the state $x = (x^1, \ldots, x^D)^\top$ is given by

$$u(x) = R_0 + 0.5 \prod_{d=1}^{D} \mathbb{I}\left[\left|x^d - 0.5\right| \in (0.25, 0.5]\right] + 2 \prod_{d=1}^{D} \mathbb{I}\left[\left|x^d - 0.5\right| \in (0.3, 0.4)\right], \quad (7)$$

where $\mathbb{I}$ is an indicator function and $R_0$ is a constant controlling the difficulty of exploration. This objective function has peaks of height $2.5 + R_0$ near the four corners of the HyperGrid, surrounded by plateaux of height $0.5 + R_0$. These plateau are situated on wide valley of height $R_0$.

**Network Struture** We use a shared encoder to parameterize the state flow estimator $F(s)$ and transition probability estimator $P_F(\cdot|s), P_B(\cdot|s)$, and one tensor to parametrize normalizing constant $Z$. The encoder is an MLP with 2 hidden layers and 256 hidden dimensions. We use ReLU as the activation function. We use Adam optimizer with a learning rate of 0.1 for $Z_\theta$'s parameters and a learning rate of 0.001 for the neural network's parameters.

### E.2.1 BACKWARD KL REGULARIZATION

In this subsection, we validate the effectiveness of backward KL regularization in Appendix E.1. We set the reward $R(x) = u(x)$ defined in Equation (7).

Following the definition of HyperGrid in Section 4.1, we consider two grids with the same number of terminal states: a 2-dimensional grid with $H = 64$ and a 4-dimensional grid with $H = 8$. We set $R_0 = 0.1, 0.01, 0.001$, where $R_0$ is defined in Equation (7). We remark that larger $H$ expects longer trajectories and smaller $R_0$ poses greater exploration challenges since models are less likely to pass the low-reward valley. We analyze training behaviors of fixed and regularized $P_B$. During the training, we update the model on actively sampled 1000 terminal states in each round, for 1000 rounds in total.

We expect that KL regularized $P_B$ converges faster than fixed $P_B$ in hard instances, such as long trajectory lengths and small $R_0$, and avoid bias introduced by simply trainable $P_B$ without regularization. To validate this, we graph the progression of the $\ell_1$ error between the target reward distribution and the observed distribution of the most recent $10^5$ visited states, and the empirical KL divergence of learned $P_B$ during training in Figure E.1, Figure E.2. We observe that fixed $P_B$ makes it very slow to converge in $\ell_1$ distance in hard instances (i.e. $R_0 \leq 0.01$ and $H = 64$), but a regularized $P_B$ with proper $\lambda_{\mathrm{KL}}$, e.g. 0.1, 1, can converge faster, and keep a near-uniform backward transition probability during training. We also observe that in short trajectories, fixed or trainable $P_B$ does not have a convergence speed difference.

To illustrate the bias of trainable $P_B$ without regularization, we visualize the learned sampler after $10^6$ states by plotting the probability of each state to perform the terminal action in Figure E.3. The initial state is $(0, 0)$, and the reward is symmetric with respect to the diagonal from $(0, 0)$ to $(63, 63)$. Therefore, the learned probability of terminal action should be symmetric with respect to the diagonal. We observe that standard TB training is biased towards the upper diagonal part, while fixed TB and regularized TB behave more symmetrically.

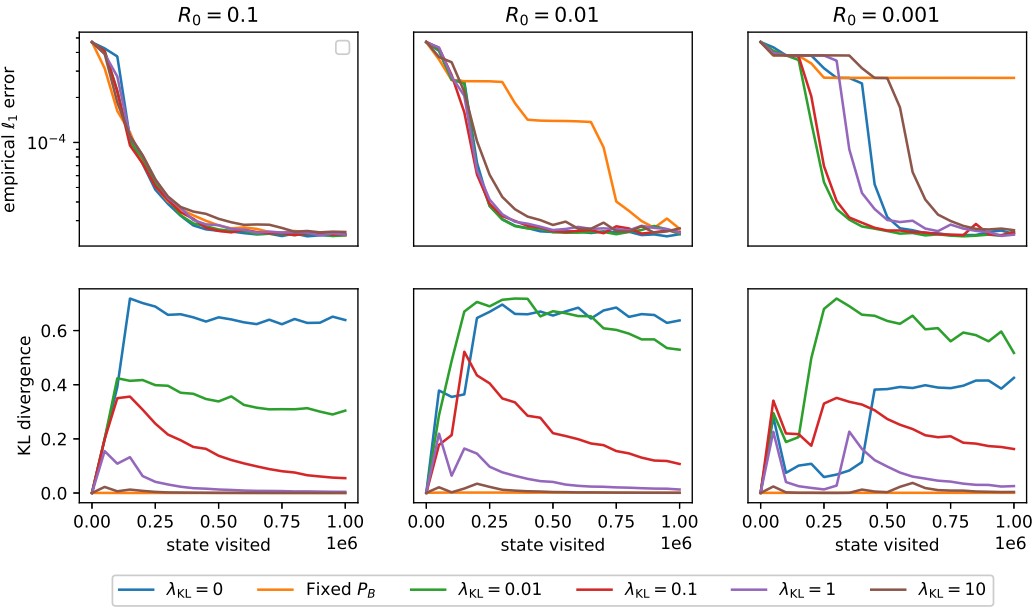

Figure E.1: **HyperGrid** with $D = 2, H = 64$, different $R_0 = 0.1, 0.01, 0.001$. $P_B$ is trainable with KL regularization weight $\lambda_{\mathrm{KL}} = 0, 0.01, 0.1, 1, 10$, or $P_B$ is fixed.

### E.2.2    Ablation study of $R_0$

In this subsection, we provide the ablation study of $R_0 = \{0, 0.0001, 0.001, 0.01, 0.1, 1\}$ for Section 4.1. cWe set $(D, H) = (2, 64), (3, 32)$ and $(4, 16)$, and compare TB and order-preserving TB (OP-TB). For $(D, H) = (2, 64)$, we plot the observed distribution on $4000$ most recently visited states in Figure E.4. We additionally plot the following three ratios: 1) #(distinctly visited states)/#(all the states); 2) #(distinctly visited maximal states)/ #(all the maximal states); 3) In the most recently $4000$ visited states, #(distinctly maximal states)/4000 in Figure E.5. We train the network for 500 steps, 200 trajectories per step, 20 steps per checkpoint.

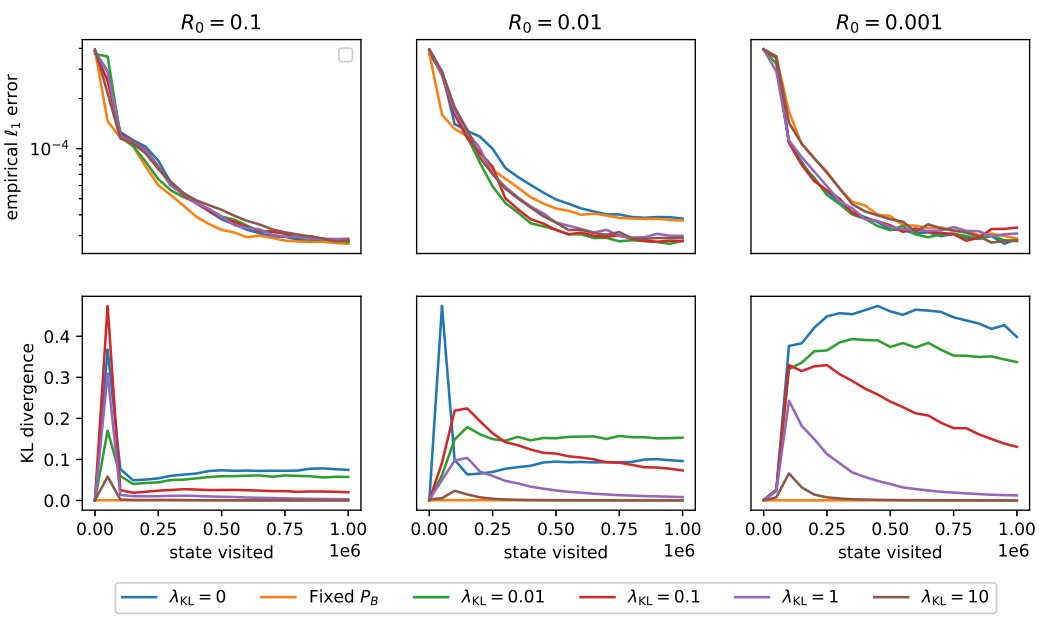

Figure E.2: **HyperGrid** with $D = 4, H = 8$, different $R_0 = 0.1, 0.01, 0.001$. $P_B$ is trainable with KL regularization weight $\lambda_{\mathrm{KL}} = 0, 0.01, 0.1, 1, 10$, or $P_B$ is fixed.

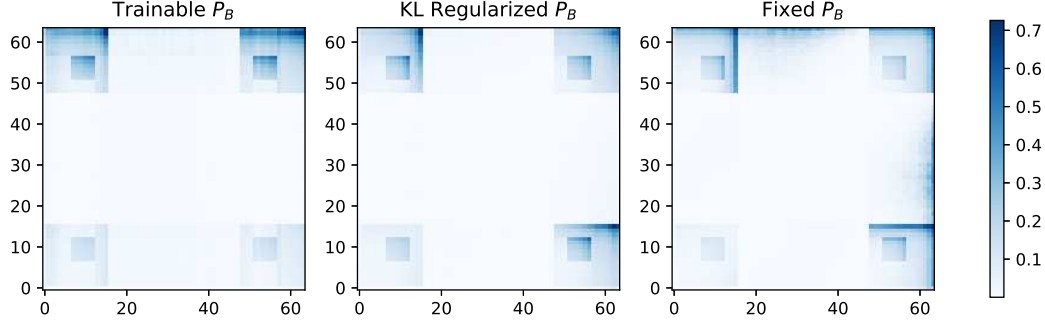

Figure E.3: **HyperGrid** with $D = 2, H = 64, R_0 = 0.01$. We set $\lambda_{\mathrm{KL}} = 1$ for KL regularization. We plot the probability of each state to perform the terminal action.

We remark that $R_0$ plays a similar role as the reward exponent $\beta$ to flatten or sparsify the rewards. Large $R_0$ facilitates exploration but hinders exploitation since a perfectly trained GFlowNet will also sample non-maximal objective candidates with high probability; whereas low $R_0$ facilitates exploitation but hinders exploration since low reward valleys hinder the discovery of maximal objective candidates far from the initial state. A good sampling algorithm should have small ratio 1), and large ratios 2), 3), which means it can sample diverse maximal states (large ratio 2), exploration), and sample only maximal states (large ratio 3), exploitation), using the fewest distinct visited states (small ratio 1), efficiency). We observe from Figure E.5 that OP-TB outperforms TB in almost every $R_0$ in terms of three ratios.

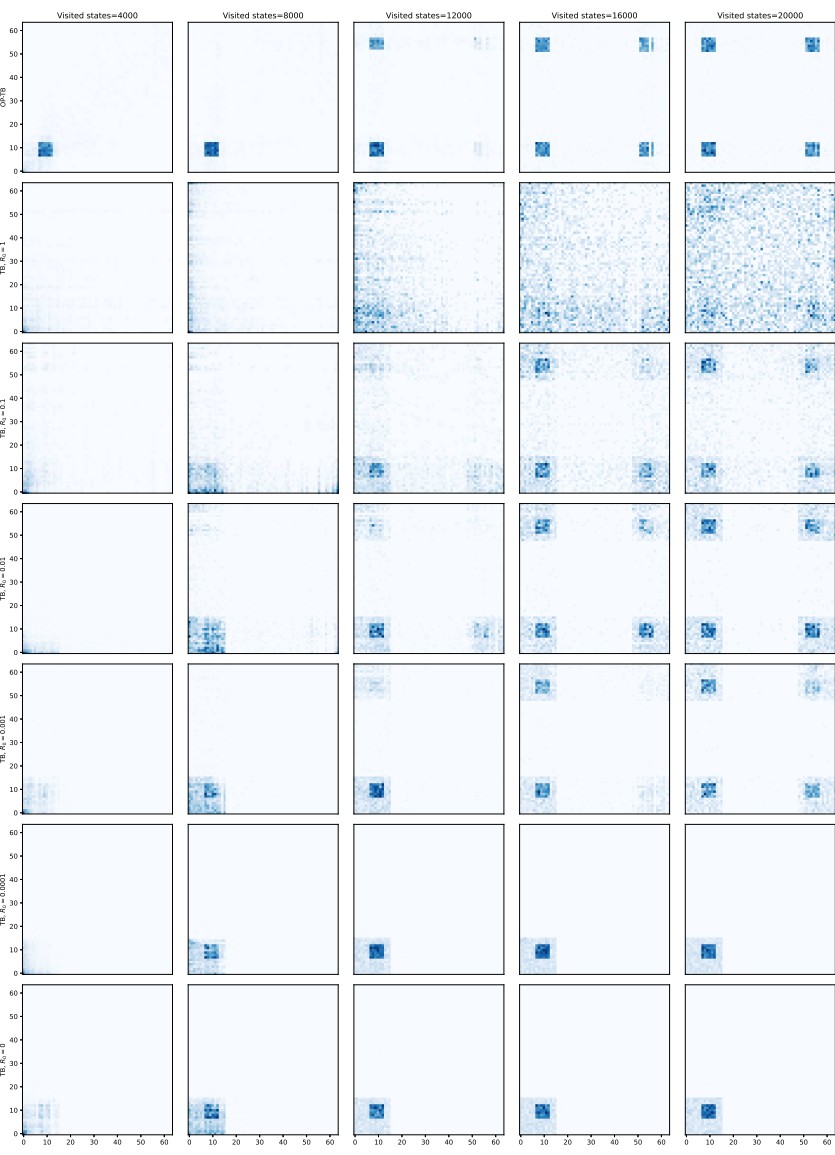

Figure E.4: Full experiments on $R_0 = 1, 0.1, 0.01, 0.001, 0.0001, 0$. We plot observed distribution on 4000 most recently visited states, when we have sampled $4000, 8000, 12000, 16000, 20000$ states.

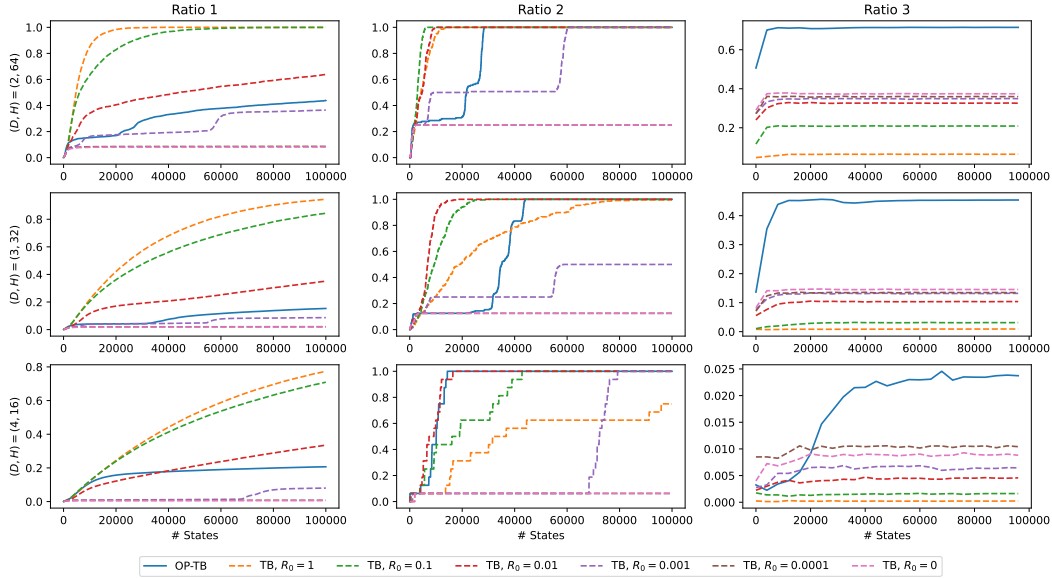

Figure E.5: Full experiments results on the HyperGrid with $(D, H) = (2, 64), (3, 32), (4, 16)$ and $R_0 = 0, 0.0001, 0.001, 0.01, 0.1, 1$. We plot the ratio 1) #(distinctly visited states)/#(all the states); 2) #(distinctly visited maximal states)/ #(all the maximal states); 3) In the most recently 4000 visited states, #(distinctly maximal states)/4000. by OP-TB (solid lines) and TB (dashed lines) method.

### E.2.3 $\widehat{R}(\cdot)$ DISTRIBUTION

We consider the cosine objective function. The objective function at the state $x = (x^1, \cdots, x^D)^\top$ is given by

$$u(x) = R_0 + \prod_{d=1}^{D} \left( \cos(50x^d) + 1 \right) (\phi(0) - \phi(5x^d)),$$

where $\phi$ is the standard normal p.d.f. Such choice offers more complex landscapes than those in Section 4.1. We set $D = 2, H = 32, R_0 = 1$ in the HyperGrid. Since DB directly parametrizes $F(s)$, we adopt the OP-DB method with $\lambda_{\text{OP}} = 0.1$. Assume there are $n$ distinct values of $u$, $u_1 < u_2 < \cdots < u_n$, among all terminal states, we calculate the mean of the learned $\widehat{R}(\cdot)$ on all the states with objective value $u_i$, i.e.,

$$\widehat{R}_i(\theta) := \text{Avg}\{\widehat{R}(x) := F(x; \theta), \text{where } u(x) = u_i\}, \quad 1 \le i \le n.$$

We sample 20 states per round and checkpoint the GFlowNet every 50 rounds. In Figure E.6, we plot the log normalized $R_i$ and $\widehat{R}_i$ by linearly scaling their sum over $i$ to 1. We observe that $\log \widehat{R}_i$ is approximately linear, confirming Propositions 1 and 2, and the slope is increasing as the training goes.

We note that the active training process we used in practice, will put more mass on states of near-maximal objective values than Propositions 1 and 2's predictions based on full batch training. In other words, we observe that the slope is larger in the near-maximal values, as explained in the following. Since we are actively training the GFlowNets, it is more likely to collect high-value states in the later training stages. Therefore, the order-preserving method in active training pays more attention to high-value states as the training goes, resulting in a larger slope, compared with log linear's prediction in Propositions 1 and 2, in the near-maximal objective value states.

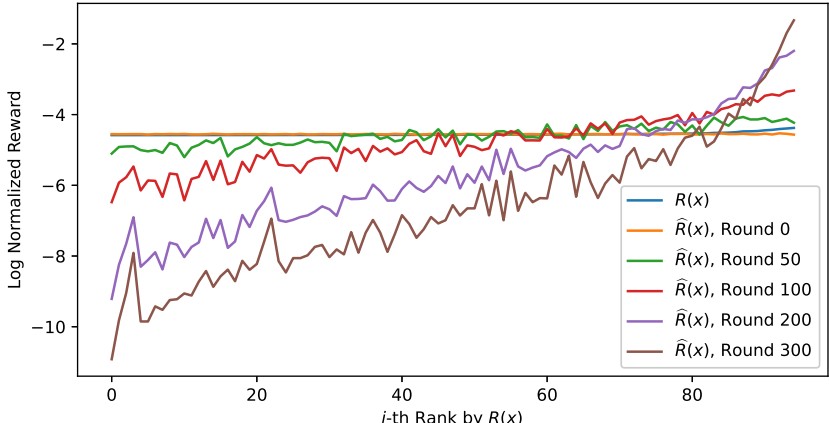

Figure E.6: **Log normalized $R(\cdot)$ (red) and $\widehat{R}(\cdot)$.** We checkpoint the GFlowNet sampler at training round 0 (randomly initialized), 50, 100, 200, 400.

### E.3 MOLECULAR DESIGN

#### E.3.1 ENVIRONMENTS

In the following, we describe each environment and its training setup individually. We adopt the sequence prepend/append MDPs as proposed by Shen et al. (2023) to simplify the graph sampling problems. For instance, in this section, we employ the sEH reward with 18 fragments and approximately $10^7$ candidates. It is worth noting that the original sEH environment (Bengio et al., 2021a) is a fragment-graph-based MDP with around 100 fragments and more than $10^{16}$ candidates. We will utilize the sEH reward with this fragment-graph-based MDP for multi-objective optimization problems in Section 5.4.

**Bag** ($|\mathcal{X}| = 96,889,010,407$) A multiset of size 13, with 7 distinct elements. The base objective value is 0.01. A bag has a "substructure" if it contains 7 or more repeats of any letter: then it has objective value 10 with 75% chance and 30 otherwise. We define the maximal objective value candidates as the $x \in \mathcal{X}$ with objective value 30, and the total number is 3160. The policy encoder is an MLP with 2 hidden layers and 16 hidden dimensions. We use an exploration epsilon $\varepsilon_F = 0.10$. The number of active training rounds is 2000.

**SIX6 (TFBind8)** ($|\mathcal{X}| = 65,536$) A string of length 8 of nucleotides. The objective function is wet-lab measured DNA binding activity to a human transcription factor, SIX6_REF_R1, from Barrera et al. (2016); Fedus et al. (2020). We define the optimal candidates as the top $0.5\%$ of $\mathcal{X}$ as ranked by the objective function, and the number is 328. The policy encoder is an MLP with 2 hidden layers and 128 hidden dimensions. We use an exploration epsilon $\varepsilon_F = 0.01$, and a reward exponent $\beta = 3$. The number of active training rounds is 2000.

**PHO4 (TFBind10)** ($|\mathcal{X}| = 1,048,576$) A string of length 10 of nucleotides. The objective function is wet-lab measured DNA binding activity to yeast transcription factor PHO4, from Barrera et al. (2016); Fedus et al. (2020). We define the optimal candidates as the top $0.5\%$ of $\mathcal{X}$ as ranked by the objective function, and the total number of $x$ is 5764. The policy encoder is an MLP with 3 hidden layers and 512 hidden dimensions. We use an exploration epsilon $\varepsilon_F = 0.01$, and a reward exponent $\beta = 3$, and scale the reward to a maximum of 10. The number of active training rounds is 20000.

**QM9** ($|\mathcal{X}| = 161,051$) A small molecule graph (Blum & Reymond, 2009; Montavon et al., 2013). The objective function is from a proxy deep neural network predicting the HOMO-LUMO gap from density functional theory. We build by 12 building blocks with 2 stems, and generate with 5 blocks per molecule. We define the optimal candidates as the top $0.5\%$ of $\mathcal{X}$ as ranked by the objective function, and the total number is 805. The policy encoder is an MLP with 2 hidden layers and 1024 hidden dimensions. We use an exploration epsilon $\varepsilon_F = 0.10$, and a reward exponent $\beta = 5$, and scale the reward to a maximum of 100. The number of active training rounds is 2000.

**sEH** ($|\mathcal{X}| = 34,012,224$) A small molecule graph. The objective function is from a proxy model trained to predict binding affinity to soluble epoxide hydrolase. Specifically, we use a gradient-boosted regressor on the graph neural network predictions from the proxy model provided by Bengio et al. (2021a) and Kim et al. (2023b), to memorize the model. Their proxy model achieved an MSE of 0.375, and a Pearson correlation of 0.905. We build by 18 building blocks with 2 stems and generate 6 blocks per molecule. We define the optimal candidates as the top $0.1\%$ of $\mathcal{X}$ as ranked by the objective function, and the total number is 34013. The policy encoder is an MLP with 2 hidden layers and 1024 hidden dimensions. We use an exploration epsilon $\varepsilon_F = 0.05$, and a reward exponent $\beta = 6$, and scale the reward to a maximum of 10. The number of active training rounds is 2000.

### E.3.2 Experimental Details

Our implementation is adapted from official implementation from Shen et al. (2023); Kim et al. (2023b).[6] We follow their environment and training settings in the codes.

**Network structure** . When training the GFNs and OP-GFNs, instead of parametrizing forward and backward transition by policy parametrization, we first parametrize the edge flow $F(s \to s'; \theta)$, and define the forward transition probability as $F(s \to s'; \theta) / \sum_{s'':s \to s'' \in \mathcal{A}} F(s \to s''; \theta)$. We clip the gradient norm to a maximum of 10.0, and the policy logit to a maximum of absolute value of 50.0. We initialize $\log Z_\theta$ to be 5.0, which is smaller than the ground-truth $Z$ in every environment. We use Adam optimizer with a learning rate of 0.01 for $Z_\theta$'s parameters and a learning rate of 0.0001 for the neural network's parameters. We do not share the policy parameters between forward and backward transition functions. When training the RL-based methods, we set the actor and critic networks' initial learning rate to be 0.0001. We set the A2C's entropy regularization parameter to be 0.01, and SQL's temperature parameter to be 0.01.

**Training and Evaluation** . In each round, we first update the model using the on-policy batch of size 32 and then perform an additional update on one off-policy batch of size 32, which was generated by the backward trajectories augmentation PRT from the replay buffer. The number of active training rounds varies for different tasks, see Appendix E.3.1 for details. To monitor the training process, for every 10 active rounds, we sample 128 monitoring candidates from the current training policy without random noise injection. At each monitoring point, we record the average of the value of the top 100 candidates ranked by the objective function, and the number of optimal candidates being found, among all the generated candidates.

### E.4 Neural Architecture Search

### E.4.1 NAS Environment

In this subsection, we study the neural architecture search environment *NATS-Bench* (Dong et al., 2021), which includes three datasets: CIFAR10, CIFAR-100, and ImageNet-16-120. We choose the *topology search space* in NATS-Bench, i.e. the densely connected DAG of 4 nodes and the operation set of 5 representative candidates. The representative operations $\mathcal{O} = \{o_k\}_{k=1}^5$ are 1) zero, 2) skip connection, 3) 1-by-1 convolution, 4) 3-by-3 convolution, and 5) 3-by-3 average pooling layer. Each architecture can be uniquely determined by a sequence $x = \{o'_{ij} \in \mathcal{O}\}_{1 \le i < j \le 4}$ of length 6, where $o'_{ij}$ indicates the operation from node $i$ to node $j$. Therefore, the neural architecture search can be regarded as an order-agnostic sequence generation problem, where the objective function of each sequence is determined by the accuracy of the corresponding architecture.

**AutoRegressive MDP Design** We use the GFlowNet $(\mathcal{S}, \mathcal{A}, \mathcal{X})$ to tackle the problem. Each state is a sequence of operations of length 6, with possible empty positions, i.e. $\mathcal{S} = \{s | s = \{o'_{ij} \in \mathcal{O} \cup \{\emptyset\}\}_{1 \le i < j \le 4}\}$. The initial state is the empty sequence, and terminal states are full sequences, i.e. $\mathcal{X} = \{x | x = \{o'_{ij} \in \mathcal{O}\}_{1 \le i < j \le 4}\}$. Each forward action fills the empty position in the non-terminal state with some $o_k$, and each backward action empties some non-empty position.

**Objective Function Design** For $x \in \mathcal{X}$, the objective function $u_T(x)$ is the test accuracy of $x$'s corresponding architecture with the weights at the $T$-th epoch during its standard training pipeline. To measure the cost to compute the objective function $u_T(x)$, we introduce the simulated train and test (T&T) time, which is defined by the time to train the architecture to the epoch $T$ and then evaluate its test accuracy. NATS-Bench provides APIs on $u_T(x)$ and its T&T time for $T \le 200$. Following the

---

[6]https://github.com/maxwshen/gflownet

experimental setups in Dong et al. (2021), when training the GFlowNet, we use the test accuracy at epoch 12 ($u_{12}$) as the objective function; when evaluating the candidates, we use the test accuracy at epoch 200 $u_{200}$ as the objective function. We remark that $u_{12}$ is a proxy for $u_{200}$ with lower T&T time, and order-preserving methods only preserve the order of $u_{12}$, ignoring the possible unnecessary information: $u_{12}$'s exact value.

### E.4.2 EXPERIMENTAL DETAILS

**Network Structure** We use the default Network structure in the package torchgfn (Lahlou et al., 2023). We use a shared encoder to parameterize the state flow estimator $F(s)$ and transition probability estimator $P_F(\cdot|s), P_B(\cdot|s)$, and one tensor to parameterize normalizing constant $Z$. The encoder is an MLP with 2 hidden layers and 256 hidden dimensions. We use ReLU as the activation function. We use Adam optimizer with a learning rate of 0.1 for $Z_\theta$'s parameters and a learning rate of 0.001 for the neural network's parameters.

**Training Pipeline** We focus on training the GFlowNet in a multi-trial sampling procedure. We find it is beneficial to use randomly generated initial datasets, and set the size to 64. In each active training round, we generate 10 new trajectories using the current training policy and update the GFlowNet on all the collected trajectories. We optionally use backward trajectory augmentation to sample 20 terminal states from the replay buffer and generate 20 trajectories per terminal using the current backward policy to update the GFlowNet.

**Multi-Trial Methods** We use the official implementation from NATS-Bench[7], where the hyperparameters are specified below.

- **REA**. We follow Algorithm 1 in Real et al. (2019). We set the number of cycles $C = 200$, the number of individuals to keep in the population $P = 10$, and the number of individuals that should participate in each tournament $S = 3$.

- **BOHB**. We follow Algorithm 2 in Falkner et al. (2018). We set the fraction of random configurations $\rho = 0$, the bandwidth factor $b_w = 3$, and the number of candidates for the acquisition function $N_s = 4$.

- **REINFORCE** (Williams, 1992). The code is directly adapted from the standard RL example [8]. We use the Adam optimizer for the policy parametrization, and set the learning rate to be 0.01. The expected reward is calculated by the exponential moving average with a momentum of 0.9.

### E.4.3 BOOSTING THE SAMPLER

Once we get a trained GFlowNet sampler, we can also use the learned order-preserving reward as a proxy to further boost it, see Appendix E.1. We adopt the following experimental settings. The (unboosted) GFlowNet samplers are obtained by training on a fixed dataset, i.e. the checkpoint sampler after the first round of the previous multi-trial training. We measure the sampler's performance by sequentially generating candidates and recording the highest validation accuracy obtained so far. We also plot each algorithm's sample efficiency gain $r_{\text{gain}}$, which indicates that the baseline (unboosted) takes $r_{\text{gain}}$ times of number of candidates to reach a target accuracy compared to that algorithm. We plot the average over 100 seeds in Figure E.7, observing that setting $r_{\text{boost}} \approx 8$ reaches up to 300% gain.

### E.4.4 ABLATION STUDY

In this subsection, we provide the ablation study of hyperparameters on GFlowNet training. To keep the comparison fair, we use the TB as the GFlowNet objective, and disable the backward trajectories augmentation in all the following experiments. We plot the test accuracy at the 12th epoch and the 200th epoch w.r.t. the estimated T&T time, following experimental settings in Section 4.3.

**OP-GFN v.s. GFN-$\beta$.** We use $R(x) := u(x)^\beta$ in TB-$\beta$ training. We set the reward exponent $\beta = 4, 8, 16, 32, 64, 128$. We disable the KL regularization in all the experiments. The results are plotted in Figure E.8. We observe that $\beta = 8 \sim 32$ are the best choices. Setting $\beta$ to be too large or small will both negatively impact the performance, by hindering exploration and exploitation

---

[7]https://github.com/D-X-Y/AutoDL-Projects/tree/main/exps/NATS-algos
[8]https://github.com/pytorch/examples/blob/master/reinforcement_learning/reinforce.py

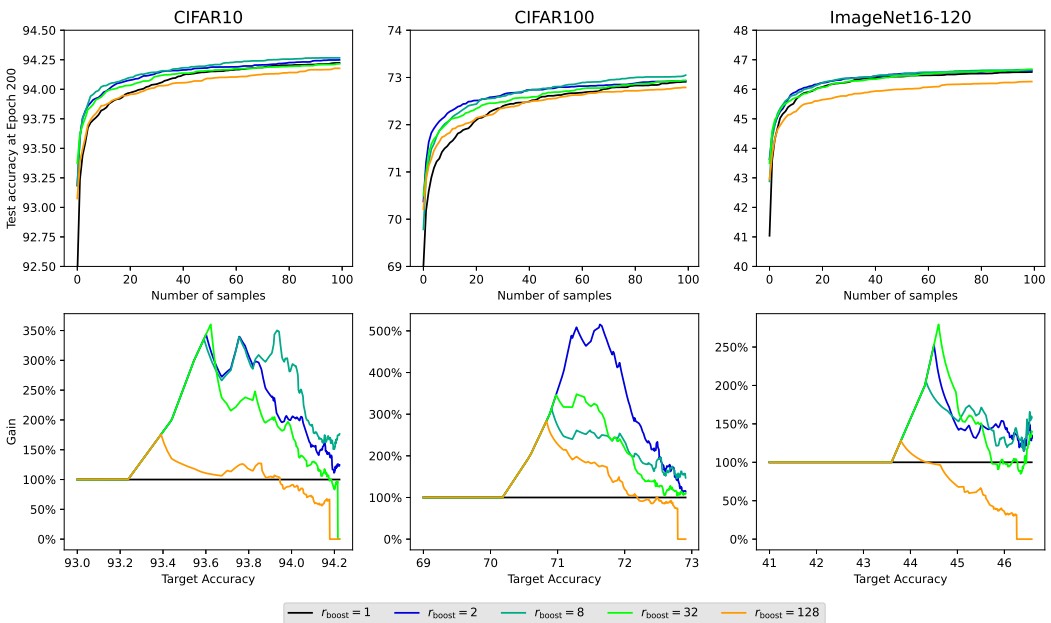

Figure E.7: **Boosting a GFlowNet sampler**. To boost the sampler, we select the best candidates ranked by $\widehat{R}(\cdot)$ from $r_{\text{boost}} = 1, 2, 8, 32, 128$ candidates states, where $r_{\text{boost}} = 1$ denotes the unboosted sampler. We plot the highest test accuracy observed so far in the 100 candidates, and the performance gain w.r.t. each target accuracy.

respectively. However, the best choices of $\beta$ are still below the performance of the OP-TB method, especially in the early training stage.

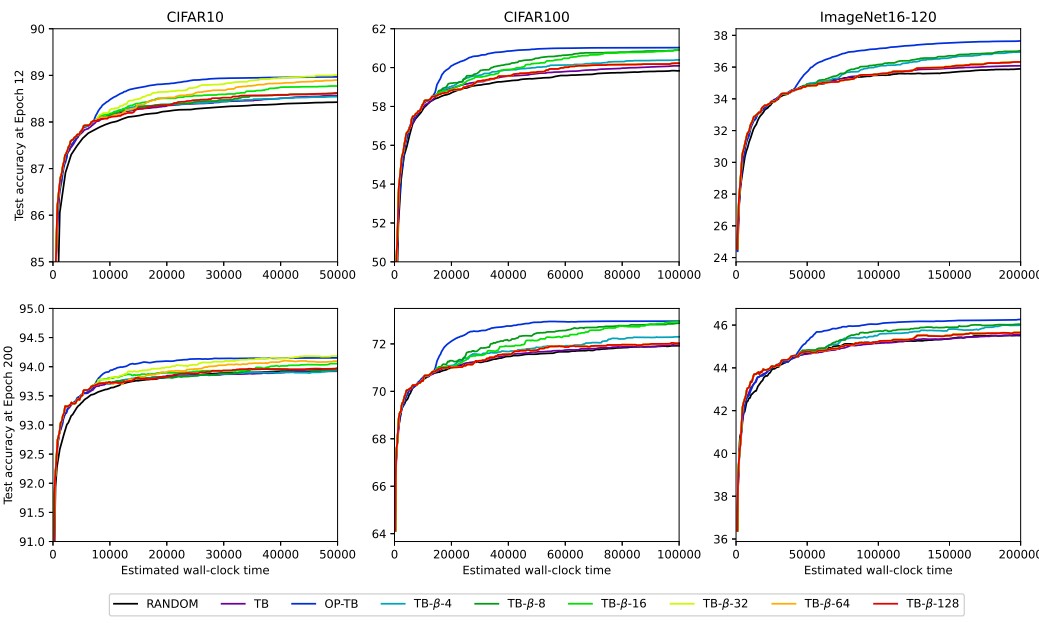

Figure E.8: **Ablation study of TB-$\beta$** with $\beta = 4, 8, 16, 32, 64, 128$. We compare them against RANDOM, TB, and OP-TB.

**Ablation Study of** $\lambda_{\text{KL}}$**.** We set the KL regularization hyperparameter $\lambda_{\text{KL}} = 0.001, 0.01, 0.1, 1$. We use the OP-TB as the OP-GFN criterion. The results are plotted in Figure E.9. We observe that a positive $\lambda_{\text{KL}}$ can contribute positively towards the sampling efficiency, but the performance is insensitive to the exact value of $\lambda_{\text{KL}}$. We empirically set $\lambda_{\text{KL}} = 0.1 \cdot \lambda_{\text{OP}}$ in default for KL regularized OP-GFN methods (recall that we set the $\lambda_{\text{OP}} = 1$ for OP-TB).

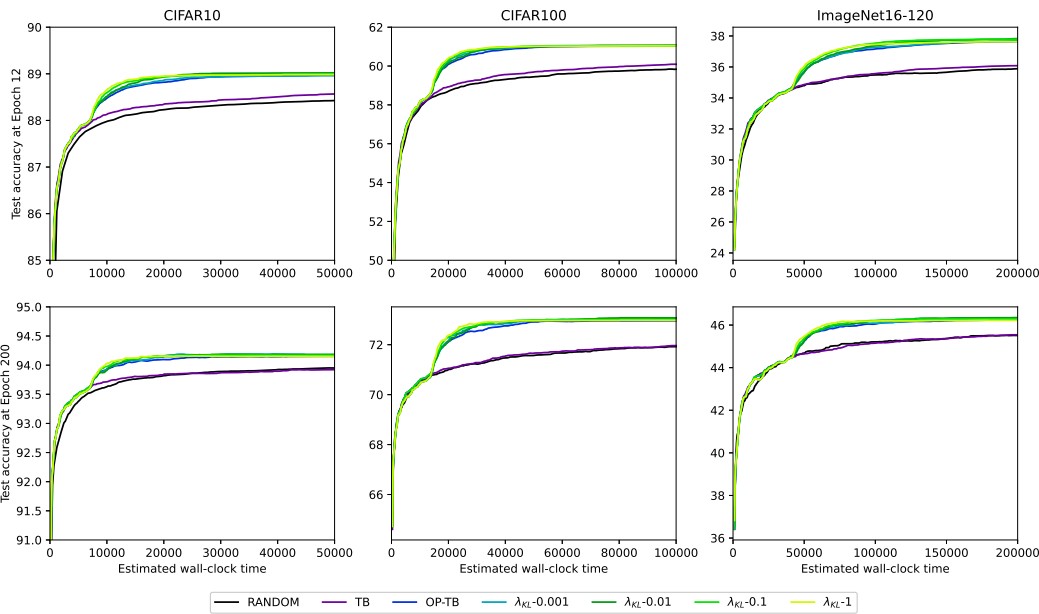

Figure E.9: **Ablation study of KL regularization** with $\lambda_{\text{KL}} = 0.001, 0.01, 0.1, 1$. We compare them against RANDOM, TB, and OP-TB without KL regularization.

**Ablation Study of** $N_{\text{init}}$**.** We set the size of the randomly generated dataset at initialization, $N_{\text{init}} = 0, 20, 40, 60, 80, 100$. We use the OP-TB as the OP-GFN criterion and disable the KL regularization in all the experiments. The results are plotted in Figure E.10. We observe that a randomly generated initial dataset contributes positively to the final performance since it avoids bias from insufficient candidates in the early training stages.

### E.4.5 OP-non-TB Methods

In this subsection, we provide the experimental results on the NAS environment by order-preserving loss on GFlowNet objectives other than TB, defined in Appendix C. Since FM does not directly parametrize $P_B$, we choose not to use the KL regularization for FM. We set $\lambda_{\text{OP}} = 0.1$, and $\lambda_{\text{KL}} = 0.01$ if the backward KL regularization is used. Other experimental settings are the same as those in Section 4.3. We include RANDOM baseline and OP-TB as comparisons. The results are plotted in Figure E.11,Figure E.12. Figure E.13. We observe that OP-non-TB methods can achieve similar performance gain with OP-TB, which validates the effectiveness and generality of order-preserving methods.

**Ablation Study of** $\lambda_{\text{OP}}$**.** We set the hyperparameter to weight the order-preserving loss $\lambda_{\text{OP}} = 0.01, 0.1, 1, 10$. We adopt OP-DB as the OP-GFN criterion and disable both the backward KL regularization and trajectory augmentation. We include the RANDOM baseline as a comparison. The results are plotted in Figure E.14. We observe that different $\lambda_{\text{OP}}$ has an influence on the performance, and $\lambda_{\text{OP}} = 0.1$ is the best choice. Therefore, similar to $\beta$, optimal $\lambda_{\text{OP}}$ is dependent on the MDP and objective function function $u(x)$. However, compared with previous ablation study on $\beta$ in Figure E.8, OP-GFN methods are less sensitive to the different choices of $\lambda_{\text{OP}}$ compared to the impact of different choices of $\beta$ on GFN-$\beta$ methods.

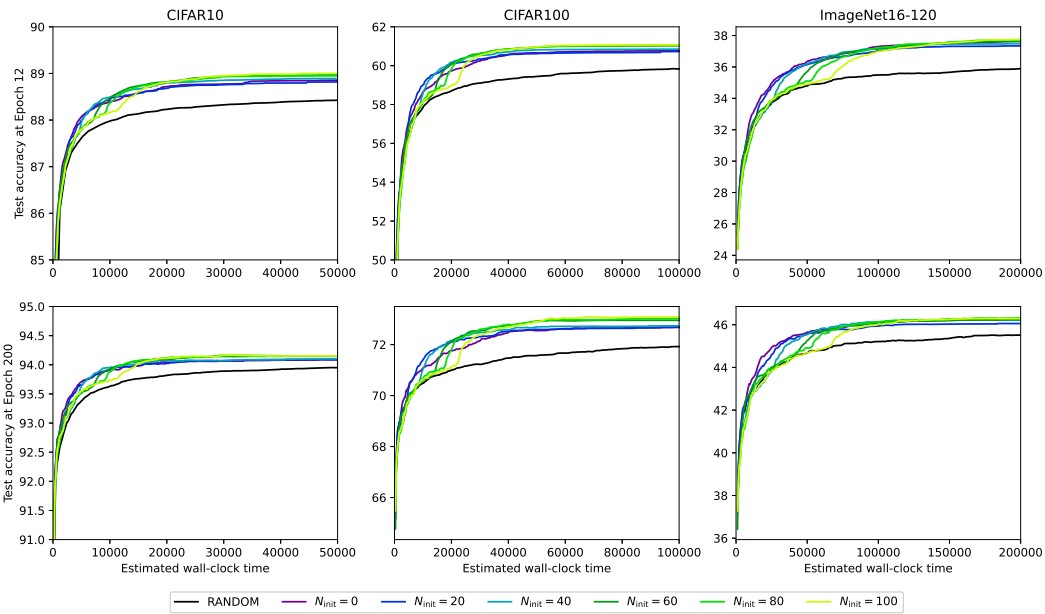

Figure E.10: **Ablation study of initial datasets** $N_{\text{init}} = 10, 20, 40, 60, 80, 100$. We compare them against RANDOM.

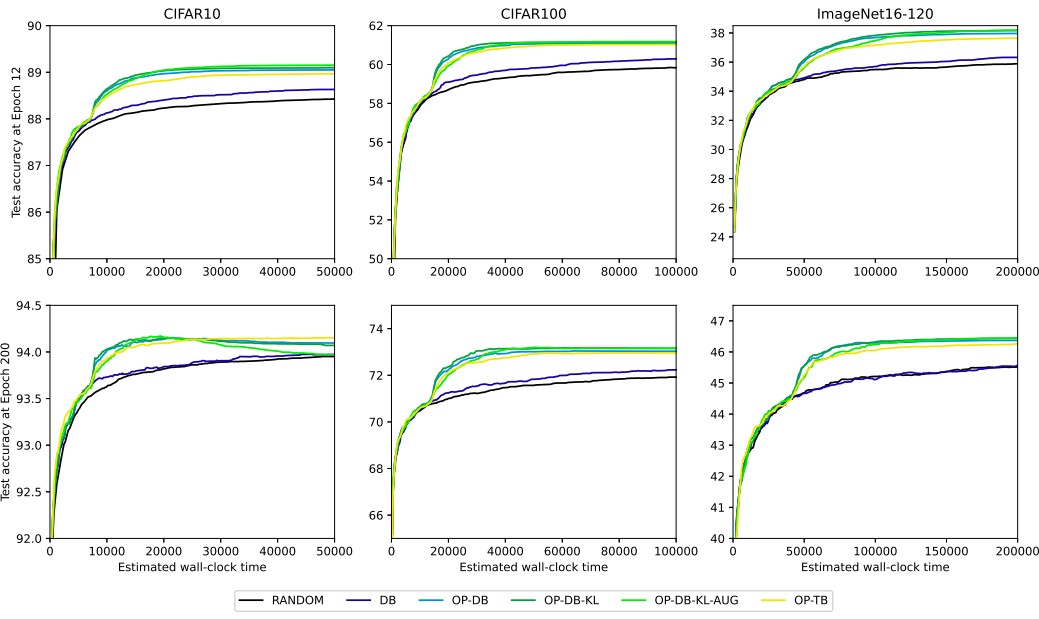

Figure E.11: **Detailed balance**. We include DB, OP-DB, OP-DB-KL, OP-DB-KL-AUG, and compare them against RANDOM and OP-TB.

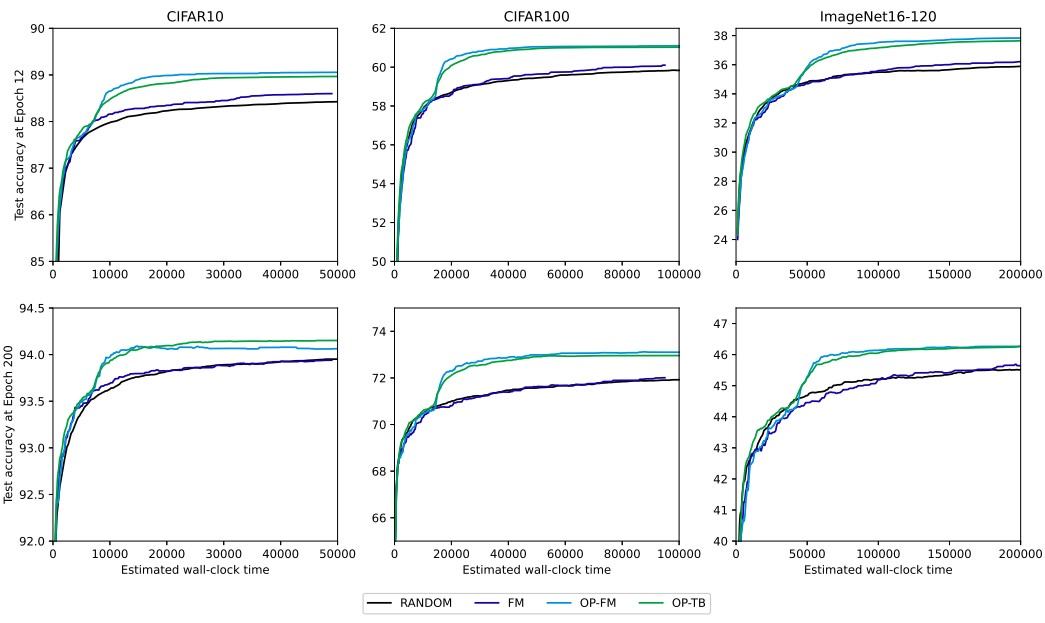

Figure E.12: **Flow matching**. We include FM, OP-FM, and compare them against RANDOM and OP-TB.

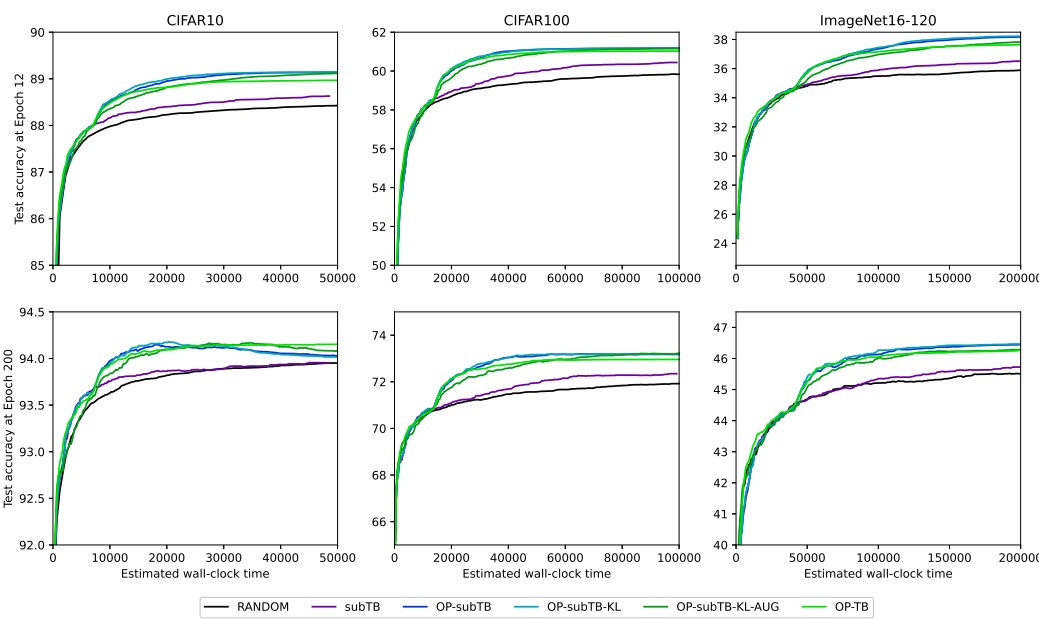

Figure E.13: **SubTrajectory balance** We include subTB, OP-subTB, OP-subTB-KL, OP-subTB-KL-AUG, and compare them against RANDOM and OP-TB.

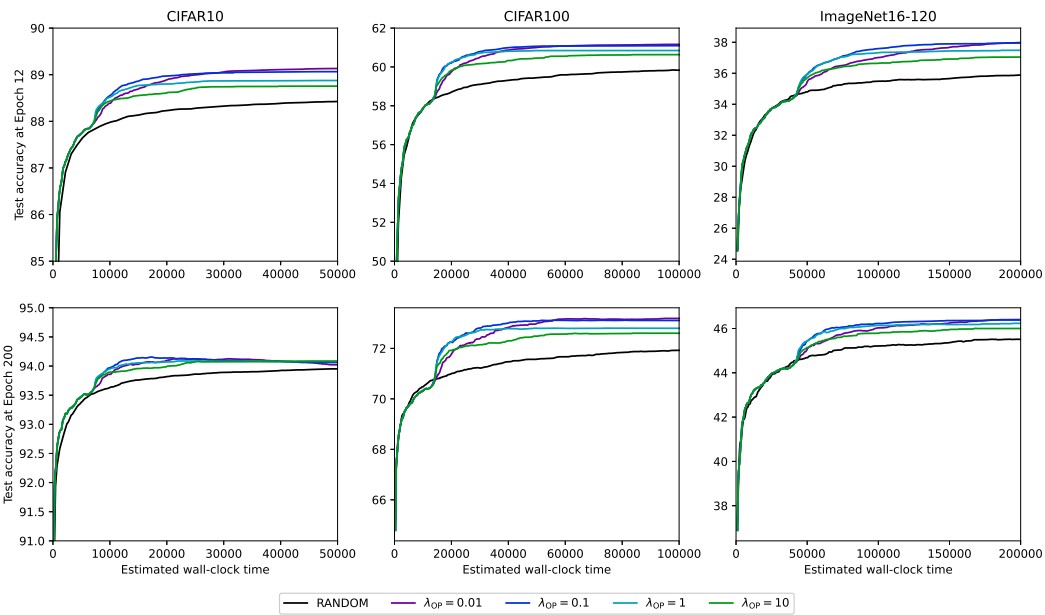

Figure E.14: **Ablation study of** $\lambda_{\text{OP}}$ with $\lambda_{\text{OP}} = 0.01, 0.1, 1, 10$. We use the OP-DB for different $\lambda_{\text{OP}}$.

## F  MULTI-OBJECTIVE EXPERIMENTS

In this section, our implementation is based on Jain et al. (2023)'s implementation and `https://github.com/recursionpharma/gflownet.git`.

### F.1  EVALUATION METRICS

We assume the reference set $P := \{\boldsymbol{p}_j\}_{j=1}^{|P|}$ is the set of the true Pareto front points, and $S = \{\boldsymbol{s}_i\}_{i=1}^{|S|}$ is the set of all the generated candidates, $P' = \text{Pareto}(S)$ is non-dominated points in $S$. When the true Pareto front is unknown, we use a discretization of the extreme faces of the objective space hypercube as $P$. We first introduce the GD class metrics:

- **Generational Distance** (GD, Van Veldhuizen (1999)) GD takes the average of the distance to the closest reference point for each generated sample: $\text{GD}(S, P) := \text{Avg}_{\boldsymbol{s} \in S} \min_{\boldsymbol{p} \in P} \|\boldsymbol{p} - \boldsymbol{s}\|_2$, which measures how close all the generated candidates are to the true Pareto front. The forward calculation of distance ensures that the indicator does not miss any part of the generated candidates.

- **Inverted Generational Distance** (IGD, Coello Coello & Reyes Sierra (2004)) IGD takes the average of the distance to the closest generated sample for each Pareto point: $\text{IGD}(S, P) := \text{Avg}_{\boldsymbol{p} \in P} \min_{\boldsymbol{s} \in S} \|\boldsymbol{p} - \boldsymbol{s}\|_2$, which measures how closely the whole Pareto front is approximated by some of generated candidates. The inverted calculation of distance ensures that the indicator does not miss any part on the Pareto front.

- **Generational Distance Plus** (GD+, Ishibuchi et al. (2015)) GD takes the average of the modified distance to the closest reference point for each generated sample: $\text{GD+}(S, P) := \text{Avg}_{\boldsymbol{s} \in S} \min_{\boldsymbol{p} \in P} \|\max(\boldsymbol{p} - \boldsymbol{s}, 0)\|_2$.

- **Inverted Generational Distance Plus** (IGD+, Ishibuchi et al. (2015)) IGD takes the average of the modified distance to the closest generated sample for each Pareto point: $\text{IGD+}(S, P) := \text{Avg}_{\boldsymbol{p} \in P} \min_{\boldsymbol{s} \in S} \|\max(\boldsymbol{p} - \boldsymbol{s}, 0)\|_2$.

- **Averaged Hausdorff distance** ($d_H$, Schutze et al. (2012)) The averaged Hausdorff distance is defined by $d_H(S, P) := \max\{\text{GD}(S, P), \text{IGD}(S, P)\}$, which combines the advantages of GD and IGD. Otherwise, if only one metric is used, candidates with small GD might

concentrate on only part of the true Pareto front, and candidates with small IGD might have high density away from the Pareto front.

Furthermore, as suggested by Ishibuchi et al. (2015), GD and IGD indicators are not Pareto compliant, which means it might contradict Pareto optimality in some cases. However, IGD+ is proved to be weakly Pareto compliant. In other words, if $S \preceq S'$ holds between two non-dominated sets in the objective space, IGD+$(S) \leq$ IGD+$(S')$ always holds, where we call a set $S$ non-dominated iff no vector in $S$ is dominated by any other vector in $S$. However, GD+ is not weakly Pareto compliant, therefore, $d_{H}+$ is not either.

**Hypervolume Indicator** (HV, Fonseca et al. (2006)), which is a standard metric reported in MOO, measuring the volume in the objective space with respect to a reference point spanned by a set of non-dominated solutions in Pareto front approximation. However, HV does not ensure uniformity in the non-convex Pareto front, the sampler might have a high HV but miss the concave part in the Pareto front.

**Pareto-Clusters Entropy** (PC-ent, Roy et al. (2023)). PC-ent measures the sampler's uniformity on the estimated Pareto front. We use the reference points $P$ and divide $P'$ in the disjoint subset $P'_j :=$ $\{\boldsymbol{p} \in P' : \forall j' \neq j, \|\boldsymbol{p} - \boldsymbol{p}_j\|_2 \leq \|\boldsymbol{p} - \boldsymbol{p}_{j'}\|_2\}$, and the entropy of the histogram is PC-ent$(P', P) :=$ $-\sum_j \frac{|P'_j|}{|P|} \log \frac{|P'_j|}{|P|}$, which attains maximum value when all the generated candidates are uniformly distributed around the true Pareto front.

$R_2$ **Indicator** (Hansen & Jaszkiewicz, 1994): $R_2$ provides a monotonic metric comparing two Pareto front approximations using a set of uniform reference vectors and a utopian point $\boldsymbol{z}^\star$ representing the ideal solution of the MOO. Specifically, we define a set of uniform reference vectors $\boldsymbol{\lambda} \in \Lambda$ that cover the space of the MOO, and a set of Pareto front approximations $\boldsymbol{s} \in S$, we calculate: $R_2(S, \Lambda, \boldsymbol{z}^\star) :=$ $\frac{1}{|\Lambda|} \sum_{\boldsymbol{\lambda} \in \Lambda} \min_{\boldsymbol{s} \in S} \max_{i \in 1, \dots, D} \{\lambda_i | z_i^* - s_i |\}$. In our experiments where all objective functions are normalized to $[0, 1]$, we set $\boldsymbol{z}^\star = (1, \cdots, 1)$. $R_2$ indicators capture both the convergence and uniformity at the same time.

## F.2 HyperGrid

We present the implementation of objectives `branin`, `currin`, `shubert`, `beale` from `https://www.sfu.ca/~ssurjano/optimization.html` in the following, where some constants, such as $308.13, 13.77$, are used for normalization to $[0, 1]$.

$$\texttt{brannin}(x_1, x_2) = 1 - \left(t_1^2 + t_2 + 10\right)/308.13,$$

where $t_1 = 15x_2 - \frac{5.1}{4\pi^2}(15x_1 - 5)^2 + \frac{5}{\pi}(15x_1 - 5) - 6, t_2 = (10 - \frac{10}{8\pi})\cos(15x_1 - 5)$;

$$\texttt{currin}(x_1, x_2) = (1 - e^{-\frac{1}{2x_2}}) \cdot \frac{2300x_1^3 + 1900x_1^2 + 2092x_1 + 60}{13.77(100x_1^3 + 500x_1^2 + 4x_1 + 20)};$$

$$\texttt{shubert}(x_1, x_2) = \left(\sum_{i=1}^{5} i\cos((i+1)x_1 + i)\right)\left(\sum_{i=1}^{5} i\cos((i+1)x_2 + i)\right)/397 + 186.8/397;$$

$$\texttt{beale}(x_1, x_2) = ((1.5 - x_1 + x_1x_2)^2 + (2.25 - x_1 + x_1x_2^2)^2 + (2.625 - x_1 + x_1x_2^3)^2)/38.8.$$

**Network Structure** For both the OP-GFNs and PC-GFNs, we fix the backward transition probability $P_B$ to be uniform and parametrize the forward transition probability $P_F$ by the MLP with 3 layers, 64 hidden dimensions and LeakyReLU as the activation function.

**Training Pipeline** During training, models are trained by the Adam optimizer (Kingma & Ba, 2014) with a learning rate of 0.01 and a batch size of 128 for 1000 steps. When training the PC-GFNs, we sample $\boldsymbol{w} \sim$ Dirichlet(1.5) and $\beta \sim \Gamma(16, 1)$. In evaluation, we sample $\boldsymbol{w} \sim$ Dirichlet(1.5) and fix $\beta = 16$. we generate 1280 candidates by the learned GFlowNet sampler as $S$, and report the metrics in Table F.1. To visualize the sampler, we plot the all the objective function vectors, and the true Pareto front, as well as the first 128 generated objective function vectors, and their estimated Pareto front, in the objective space $[0, 1]^2$ and $[0, 1]^3$ in Figure F.1. We observe that our sampler achieve better approximation (i.e. almost zero IGD+ and smaller $d_H$), and uniformity (i.e. higher PC-ent) of the Pareto front. We also plot the learned reward distributions of OP-GFNs and compare them with the indicator functions of the true Pareto front solutions in Figure 5.1, where we observe that

OP-GFNs can learn a highly sparse reward function that concentrates on the Pareto solutions. We also plot the learned reward distribution of PC-GFNs where the preference $\boldsymbol{w} \sim \mathrm{Dirichlet}(1.5)$ in Figure 5.1, where we find that PC-GFNs might miss some solutions and does not concentrate on the correct solutions very accurately.

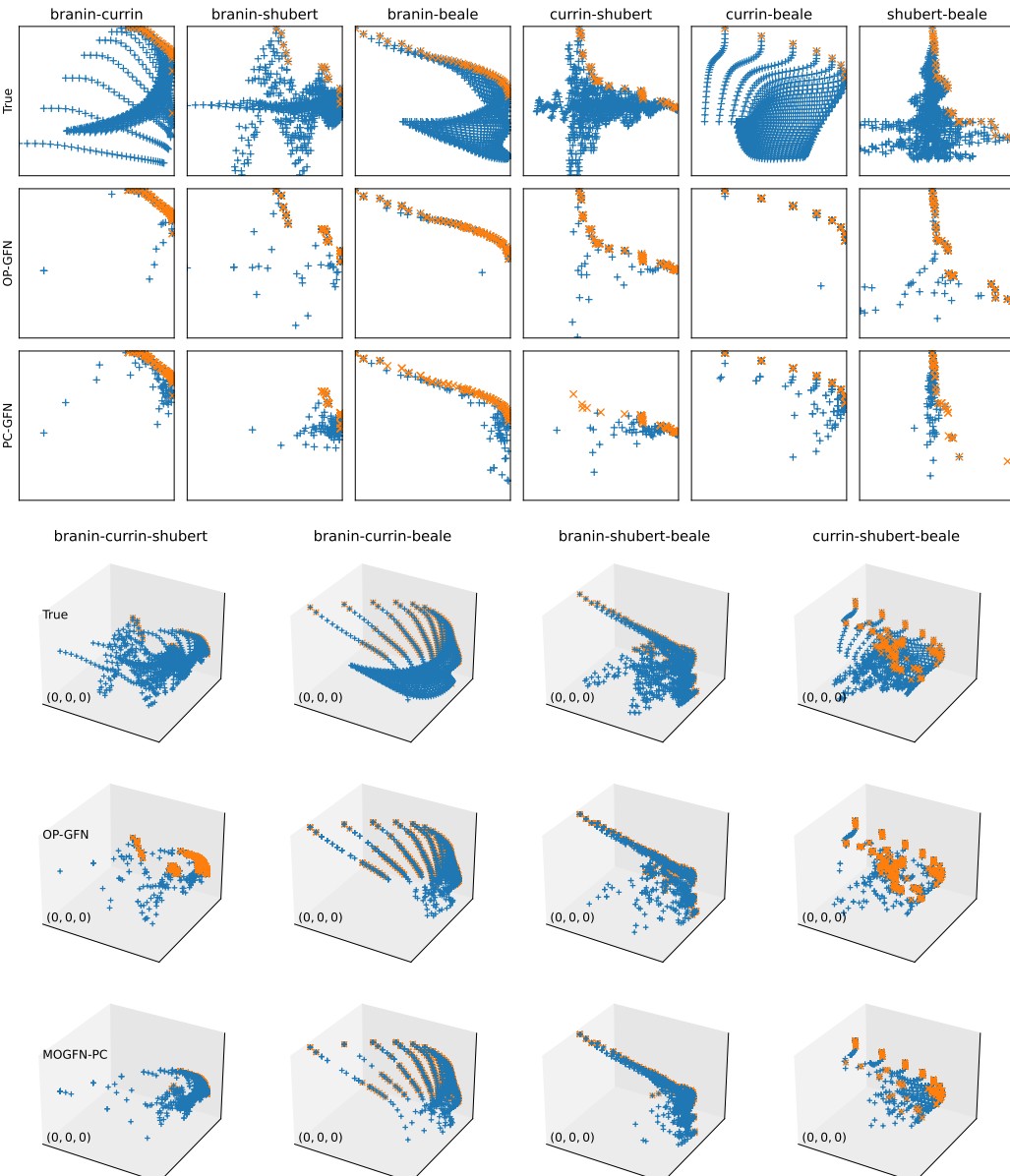

Figure F.1: **Reward Landscape**: The first row of the above two figures contains all the states (blue) and the true Pareto front (orange). In the second and third rows, we plot the first 128 generated candidates (blue) and the estimated Pareto front (orange). The $x$-, $y$-axis, (and $z$-axis) are the first, second (and third) objectives in the title of respectively.

### F.3 N-GRAMS

**Network Structure** For both the OP-GFNs and PC-GFNs, we fix the backward transition probability $P_B$ to be uniform, and parametrize the forward transition probability $P_F(\cdot|s, w)$ by the Transformer

| Objective | OP-GFN(Ours) | | | PC-GFN | | |
|---|---|---|---|---|---|---|
| | $d_H(\downarrow)$ | IGD+($\downarrow$) | PC-ent($\uparrow$) | $d_H(\downarrow)$ | IGD+($\downarrow$) | PC-ent($\uparrow$) |
| `branin-currin` | **0.018** | **2.66e-6** | **3.56** | 0.047 | 2.66e-6 | 3.49 |
| `branin-shubert` | **0.046** | **6.82e-6** | **2.56** | 0.080 | 0.055 | 2.08 |
| `branin-beale` | **0.0068** | **1.38e-8** | **3.75** | 0.064 | 1.38e-8 | 3.64 |
| `currin-shubert` | **0.039** | **1.28e-8** | **3.44** | 0.040 | 0.016 | 2.73 |
| `currin-beale` | **0.0028** | **1.56e-8** | **2.08** | 0.075 | 1.56e-8 | 2.04 |
| `shubert-beale` | **0.032** | **7.06e-9** | **3.06** | 0.034 | 0.016 | 2.21 |
| `branin-currin-shubert` | **0.032** | **9.07e-7** | **4.57** | 0.05 | 0.016 | 4.32 |
| `branin-currin-beale` | **0.013** | 1.40e-4 | **5.28** | 0.015 | **0.0010** | 5.04 |
| `branin-shubert-beale` | **0.023** | **1.44e-8** | **4.86** | 0.045 | 0.034 | 4.48 |
| `currin-shubert-beale` | **0.026** | **1.41e-8** | **4.4** | 0.048 | 0.029 | 3.58 |
| All Objectives | 0.020 | **1.77e-8** | **5.91** | **0.019** | 0.010 | 5.79 |

Table F.1: **HyperGrid Task**: $d_H(S, P)$, IGD+$(P', P)$ and PC-ent$(P', P)$ of all generated candidates $S$ and generated Pareto front $P'$ w.r.t. the true Pareto front $P$.

encoder. The Transformer encoder consists of 3 hidden layers of dimension 64 and 8 attention heads to embed the current state s. We encode the preference $w \sim$ Dirichlet$(1.0)$ by thermometer encoding with 50 bins and set reward exponent $\beta = 96$ in the PC-GFNs.

**Training Pipeline** We use the trajectory balance criterion in training the GFlowNets. During training, models are trained by the Adam optimizer (Kingma & Ba, 2014) with learning rate $10^{-4}$ for $P_F$'s parameters and $10^{-3}$ for $Z$'s parameters, and the batch size 128 for 10000 steps. In the training, we inject the random noise with probability $0.01$ in the $P_F$. In the OP-GFNs, We set the sampling temperature to be 0.1 during training and evaluation. We additionally use a replay buffer of a maximal size of 100000 and add 1000 warm-up candidates to the buffer at initialization. Instead of the on-policy update, we push the online sampled batch into the replay buffer and immediately sample a batch of the same size.

**Evaluation** To evaluate the PC-GFNs, Jain et al. (2023) samples $N$ candidates per test preference and then pick the top-$k$ candidates with highest scalarized rewards. However, in our condition-free GFlowNets, there is no scalarization. In order to align the evaluation setting, we disable the selection process of top-$k$ candidates and take all the generated candidates into account. In other words, we sample $N = 128$ candidates per $w \sim$ Dirichlet$(1.0)$ for 10 rounds, resulting in 1280 candidates in total.

| # Objectives | Unigrams | | | Bigrams | | | |
|---|---|---|---|---|---|---|---|
| 2 | `"A"`, `"C"` | | | `"AC"`, `"CV"` | | | |
| 3 | `"A"`, `"C"`, `"V"` | | | `"AC"`, `"CV"`, `"VA"` | | | |
| 4 | `"A"`, `"C"`, `"V"`, `"W"` | | | `"AC"`, `"CV"`, `"VA"`, `"AW"` | | | |

Table F.2: Objectives considered for the $n$-grams task

### F.4 DNA SEQUENCE GENERATION

We use the same experimental settings as Appendix F.3, except for setting $\beta = 80$. We report the metrics in Table F.4, and plot the objective vectors in Figure F.2. We conclude that OP-GFNs achieve similar or better performance than preference conditioning. From Figure F.2, we notice that since $\beta = 80$ is large, it samples more closely to the Pareto front (in the second column), but lacks exploration (in the first and third column).

### F.5 FRAGMENT BASED MOLECULE GENERATION

**Network Structure** We fix the backward transition probability $P_B$ to be uniform, and parametrize the forward transition probability $P_F$ by the GNN with 2 layers, the node embedding size of 64. We encode the preference vector $w$ or goal conditioning vector $d_g$ by thermometer encoding of 16 bins, and the temperature $\beta$ by thermometer encoding of 32 bins.

| Algorithm | 2 Bigrams | | | |
|---|---|---|---|---|
| | HV ($\uparrow$) | $R_2$ ($\downarrow$) | PC-ent ($\uparrow$) | Diversity ($\uparrow$) |
| PC-GFN | $0.50 \pm 0.03$ | $\mathbf{1.45 \pm 0.05}$ | $2.12 \pm 0.09$ | $\mathbf{14.96 \pm 0.27}$ |
| OP-GFN | $\mathbf{0.56 \pm 0.05}$ | $1.51 \pm 0.26$ | $\mathbf{2.38 \pm 0.12}$ | $7.56 \pm 0.27$ |
| | 2 Unigrams | | | |
| PC-GFN | $0.42 \pm 0.05$ | $1.46 \pm 0.03$ | $2.22 \pm 0.01$ | $2.97 \pm 0.31$ |
| OP-GFN | $\mathbf{0.47 \pm 0.00}$ | $1.45 \pm 0.01$ | $\mathbf{2.24 \pm 0.00}$ | $\mathbf{7.30 \pm 0.04}$ |
| | 3 Bigrams | | | |
| PC-GFN | $0.30 \pm 0.02$ | $\mathbf{8.49 \pm 0.46}$ | $\mathbf{2.54 \pm 0.11}$ | $\mathbf{14.43 \pm 0.69}$ |
| OP-GFN | $0.30 \pm 0.02$ | $9.95 \pm 0.64$ | $1.46 \pm 0.63$ | $7.37 \pm 0.14$ |
| | 3 Unigrams | | | |
| PC-GFN | $0.03 \pm 0.01$ | $10.68 \pm 0.58$ | $2.77 \pm 0.29$ | $3.86 \pm 1.17$ |
| OP-GFN | $\mathbf{0.14 \pm 0.00}$ | $10.18 \pm 0.53$ | $\mathbf{3.77 \pm 0.08}$ | $\mathbf{9.89 \pm 0.18}$ |
| | 4 Bigrams | | | |
| PC-GFN | $0.04 \pm 0.006$ | $46.58 \pm 1.47$ | $3.74 \pm 0.23$ | $15.35 \pm 0.27$ |
| OP-GFN | $0.05 \pm 0.00$ | $46.89 \pm 1.36$ | $\mathbf{4.18 \pm 0.10}$ | $\mathbf{19.77 \pm 1.01}$ |
| | 4 Unigrams | | | |
| PC-GFN | $0.005 \pm 0.00$ | $49.37 \pm 0.04$ | $3.52 \pm 0.15$ | $4.56 \pm 0.41$ |
| OP-GFN | $\mathbf{0.03 \pm 0.00}$ | $50.78 \pm 2.24$ | $\mathbf{4.87 \pm 0.12}$ | $\mathbf{11.57 \pm 0.37}$ |

Table F.3: $n$-**Grams Task:** Hypervolume, $R_2$ indicator, PC-entropy, and TopK diversity of OP-GFNs and PC-GFNs on various tasks.

| Objectives | Algorithm | HV ($\uparrow$) | $R_2$ ($\downarrow$) | PC-ent ($\uparrow$) | Diversity ($\uparrow$) |
|---|---|---|---|---|---|
| (1)-(2) | PC-GFN | 0.100 | 4.32 | 0.58 | 5.03 |
| | OP GFN | **0.122** | **3.93** | **0.73** | **9.60** |
| (1)-(3) | PC-GFN | 0.352 | 2.65 | 0.0 | 5.50 |
| | OP GFN | 0.352 | 2.65 | 0.0 | **13.70** |
| (2)-(3) | PC-GFN | 0.195 | 3.41 | 0.04 | **6.59** |
| | OP GFN | **0.25** | **2.98** | **0.77** | 3.02 |

Table F.4: **DNA sequence generation**: Hypervolume, $R_2$ indicator, PC-entropy, and TopK diversity of OP-GFN and PC-GFN on various tasks.

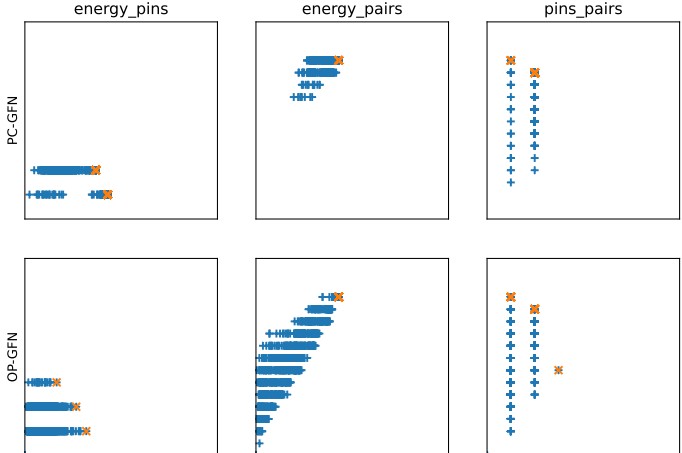

Figure F.2: **DNA sequence generation**: We plot the generated 1280 candidates (blue) and the estimated Pareto front (orange). The $x$-, $y$-axis are the first, and second objective in the title of respectively.

**Training Pipeline** We use the trajectory balance criterion in training the GFlowNets. During training, models are trained by the Adam optimizer (Kingma & Ba, 2014) with learning rate $10^{-4}$ for $P_F$'s parameters and $10^{-3}$ for $Z$'s parameters, and the batch size 64 for 20000 steps. In the training, we inject the random noise with probability 0.01 in the $P_F$. For goal conditioning and order-preserving GFlowNets, we use a replay buffer of size 100000, and 1000 warmup samples, and perform resampling. To prevent the sampling distribution from changing too abruptly, we collect new trajectories from a sampling model $P_F(\cdot|\theta_{\text{sampling}})$ which uses a soft update with hyperparameter $\tau$ to track the learned GFN at the $k$-th update: $\theta_{\text{sample}}^k \leftarrow \tau \cdot \theta_{\text{sample}}^{k-1} + (1 - \tau) \cdot \theta^k$, where $\theta^k$ is the current parameters from the learning, and we set $\tau = 0.95$.

**Algorithm Specification** : In the preference conditioning GFlowNets, we sample the preference vector $\boldsymbol{w} \sim \text{Dirichlet}(1.0)$. In the goal conditioning GFlowNets, a sample $x$ is defined in the *focus region* $g$ if the cosine similarity between its objective vector $\boldsymbol{u}$ and the goal direction $\boldsymbol{d}_g$ is above the threshold $c_g$, i.e. $g := \{\boldsymbol{u} \in \mathbb{R}^k : \cos\langle\boldsymbol{u}, \boldsymbol{d}_g\rangle := \frac{\boldsymbol{u} \cdot \boldsymbol{d}_g}{\|\boldsymbol{u}\| \cdot \|\boldsymbol{d}_g\|} \geq c_g\}$. The reward function $R_g$ depends on the current goal $g$, we set the reward to be

$$R_g(x) := \alpha_g(x) \cdot \mathbb{1}^\top \boldsymbol{u}(x) \cdot \mathbf{1}[\boldsymbol{u}(x) \in g], \quad \text{where } \alpha_g(x) = (\cos\langle\boldsymbol{u}(x), \boldsymbol{d}_g\rangle)^{\frac{\log m_g}{\log c_g}} .$$

We set the focus region cosine similarity threshold $c_g$ to be 0.98, and the limit reward coefficient $m_g$ to be 0.2. We sample the goal conditioning vector $\boldsymbol{d}_g$ from the tabular goal-sampler Roy et al. (2023). In the predefined reward $R_{\boldsymbol{w}}$ and $R_{\boldsymbol{d}_g}$, we set the reward exponent $\beta = 60$.

**Evaluation** We sample 128 candidates per round, 50 rounds using the trained sampler. We report the metrics in the following:

| Objectives | PC-GFN | | | PC-GFN | | | OP-GFN (Ours) | | |
|---|---|---|---|---|---|---|---|---|---|
| | $d_H(\downarrow)$ | IGD+($\downarrow$) | PC-ent($\uparrow$) | $d_H(\downarrow)$ | IGD+($\downarrow$) | PC-ent($\uparrow$) | $d_H(\downarrow)$ | IGD+($\downarrow$) | PC-ent($\uparrow$) |
| seh-qed | 0.56 | 0.20 | 1.88 | 0.55 | 0.20 | 1.84 | 0.55 | 0.20 | 1.87 |
| seh-sa | 0.33 | 0.15 | 1.71 | 0.34 | 0.14 | 1.58 | 0.33 | 0.15 | **2.18** |
| seh-mw | 0.14 | 0.15 | 1.33 | 0.14 | 0.14 | 1.56 | 0.14 | 0.13 | **1.69** |
| qed-sa | 0.35 | 0.06 | **1.73** | 0.34 | 0.05 | 1.47 | 0.33 | 0.07 | 1.52 |
| qed-mw | 0.16 | 0.03 | 0.00 | 0.15 | 0.03 | 0.00 | 0.15 | 0.04 | 0.00 |
| sa-mw | 0.13 | 0.00 | 0.00 | 0.14 | 0.00 | 0.00 | 0.13 | 0.00 | 0.00 |

Table F.5: **Fragment Based Molecule Generation**: $d_H(S, P)$, IGD+($P', P$) and PC-ent($P', P$) of all generated candidates $S$ and generated Pareto front $P'$ w.r.t. the true Pareto front $P$.

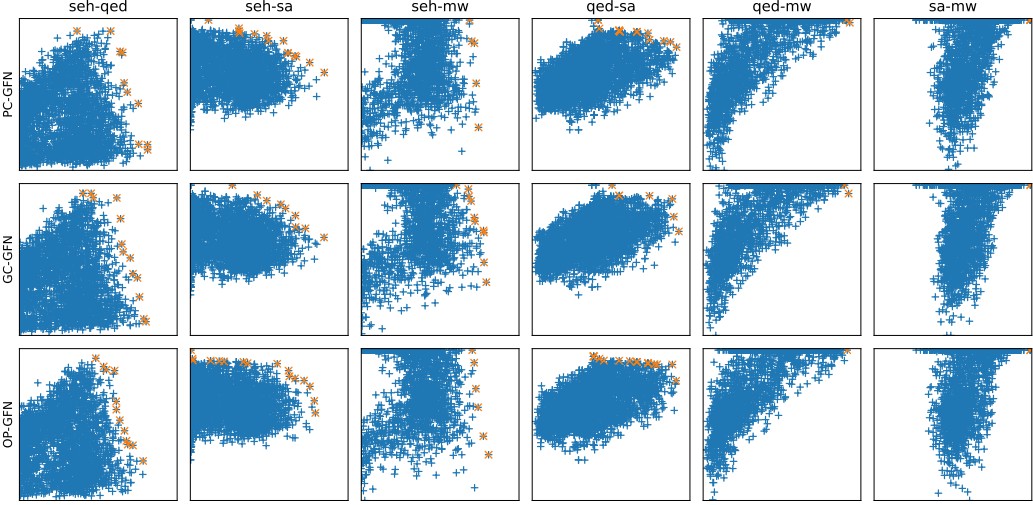

Figure F.3: **Fragment Based Molecule Generation (Full)**: We plot the generated 3200 candidates (blue) and the estimated Pareto front (orange). The $x$-, $y$-axis are the first, and second objectives in the title of respectively.

