# OpenReview forum: "Order-Preserving GFlowNets"
_ICLR.cc/2024/Conference — ICLR 2024 poster_

### Official Review · Reviewer_CCax · 2023-10-29

**Soundness:** 3 good
**Presentation:** 3 good
**Contribution:** 2 fair
**Rating:** 6
**Confidence:** 4

**Summary:**

A new method called Order-Preserving GFlowNets (OP-GFNs) has been proposed to address issues with Generative Flow Networks (GFlowNets) in sampling candidates with probabilities proportional to a given reward. GFlowNets can only be used with a predefined scalar reward, which can be computationally expensive or not directly accessible, and the conventional practice of raising the reward to a higher exponent may vary across environments. OP-GFNs eliminate the need for an explicit formulation of the reward function by sampling with probabilities in proportion to a learned reward function that is consistent with a provided (partial) order on the candidates. The training process of OP-GFNs gradually sparsifies the learned reward landscape in single-objective maximization tasks, concentrating on candidates of a higher hierarchy in the ordering to ensure exploration at the beginning and exploitation towards the end of the training. OP-GFNs demonstrate state-of-the-art performance in both single-objective maximization and multi-objective Pareto front approximation tasks.

**Strengths:**

The paper studies an interesting problem of GFlowNets -- training an order-preserving GFlowNets. The paper is also well-written and easy to follow.

**Weaknesses:**

> Note, that the optimal $\beta$ heavily depends on the geometric landscape of $u(x)$.

There have been a few related works that attempt to determine the optimal $\beta$ either from the perspective of dynamically annealing temperature or training a temeperature-conditioned GFlowNet that can generalize across a set of different temperatures.

In addition, it would be better to also compare OP-GFN with these baselines besides GFN-$\beta$.

Kim, Minsu, Joohwan Ko, Dinghuai Zhang, Ling Pan, Taeyoung Yun, Woochang Kim, Jinkyoo Park, and Yoshua Bengio. "Learning to Scale Logits for Temperature-Conditional GFlowNets." *arXiv preprint arXiv:2310.02823* (2023).

Zhang, David W., Corrado Rainone, Markus Peschl, and Roberto Bondesan. "Robust Scheduling with GFlowNets." In *The Eleventh International Conference on Learning Representations*. 2022.

> Therefore, we only focus on evaluating the GFlowNet’s ability to discover the maximal objective.

If the goal is to discover the maximal objective, why do you use GFlowNets and not RL, wher the latter learns an optimal policy that maximizes the return.

>  The ability of standard GFlowNets, i.e. $R(x) = u(x)$, is sensitive to $R_0$.

The performance of trajectory balance is indeed senstive to the value of $R_0$ (which is also mentioned in the paper), which leads to large variance. However, this is usually not the case for flow matching, detailed balance, and sub-trajectory balance. It would be better to make this claim in a more clear way. In addition, it would be better to include a more extensive study of OP-GFN with other more advanced learning objectives (e.g., FM, DB, SubTB with temporal difference based objective) with a more thorough discussion.

> Large $R_0$ encourages exploration but hinders exploitation since a perfectly trained GFlowNet will also sample non-optimal candidates with high probability; whereas low $R_0$ encourages exploitation but hinders exploration since low reward valleys hinder the discovery of maximal reward areas far from the initial state.

$R_0$ actually determines the sparsity of the reward function -- with a larger $R_0$, the reward function is less sparse, while a smaller $R_0$ corresponds to a high sparsity in the reward function. Therefore, it is inappropriate to say that "Large $R_0$ encourages exploration" and "low $R_0$" encourages exploitataion. Whether encouraging exploration (or exploitation) for the agent is determined by the learning algorithm itself, instead of $R_0$ (which is the underlying environment).

> We observe that the OP-GFN outperforms the GFN-β method by a significant margin.

It would be better to also include more advanced baselines and methods in the field of NAS.

**Questions:**

Please refer to the weaknesses part in the previous section.

---

> ### Author Response · Authors · 2023-11-14
> **Response to Reviewer CCax**
>
> 1. **Related work**. Please see the **Common Response: Related Works**.
>
> 2. **RL vs OP-GFNs**. Please see the **Common Response: Extensive Experiments \& RL vs OP-GFNs**.
>
> 3. **More baselines**. We think we mainly use the HyperGrid for sanity checking instead of sweeping experiments to compare performances, so we only consider TB here. Later, in the NAS environment, we have implemented and compared OP-TB with various baselines, including FM, DB, subTB. Please see Section E.4.5. To conclude, OP has significant improvements on all the methods. We are currently running additional experiments for the molecular design environments, which we will add as new baselines to the updated manuscript. Please see the **Common Response: Extensive Experiments**.
>
> 4. **$R_0$ and exploration-exploitation**. Sorry for the confusion. What we wanted to explain is that a high sparsity in the reward function is not good for GFNs to explore different modes. Therefore, small $R_0$ is a harder case for traditional GFN methods to discover the mode on the upper-right corner, see Figure E.4 for details. However, since the OP-GFN only focuses on the ordering, it is independent of $R_0$. We change the term "encourages" to "facilitates".
>
> 5. **Advanced baselines in NAS**. Our OP-GFNs are multi-trial sampling methods. In the NAS environments, we have compared our results against three classes multi-trial sampling methods. 1) evolutionary strategy, e.g., REA; 2) reinforcement learning (RL)-based methods, e.g., REINFORCE, 3) HPO methods, e.g., BOHB. We think these three algorithms is representative of previous baselines. We also include other GFN methods, including (OP-)DB, (OP-)FM, and (OP-)subTB. However, for the widely-used optimization-based methods, such as DARTS, since they are not multi-trial sampling, belonging to a different class of methods from OP-GFNs, we do not include the comparison.

---

> ### Author Response · Authors · 2023-11-22
> **Any further comments from the Reviewer CCax?**
>
> We appreciate your time and effort in reviewing our paper. Given the upcoming deadline for the end of the discussion period, we are wondering whether the added experiments and our responses have adequately addressed your concerns. We are willing to make more improvements based on your further comments.

---

> ### Author Response · Authors · 2023-11-23
> **Friendly reminder: The Discussion Period is ending in fewer than 12 hours.**
>
> Dear Reviewer CCax,
>
> We appreciate your time and effort in reviewing our paper. Since we have a limited 12-hour window for the author-reviewer discussion, we kindly ask for your valuable feedback on our rebuttal. We are willing to make further improvements based on your feedback.
>
> Best,
> Authors.

---

> ### Comment · Reviewer_CCax · 2023-11-23
>
> I would like to thank the authors for the additional experiments, which have clarified my concerns.
>
> Regarding Point 3 (Later, in the NAS environment, we have implemented and compared OP-TB with various baselines, including FM, DB, subTB.), I found the results and the conclusion that "We obeserve that non-OP-TB methods can achieve similar or better performance with OP-TB, which validate the effectiveness and generality of order-preserving methods." very interesting, and better support that the OP method is general, which could further improve the paper. Please incorporate these results in the final version for a more in-depth comparison.
>
> I have therefore raised my score.

---

> > ### Author Response · Authors · 2023-11-23
> > **Response to Reviewer CCax**
> >
> > We are thankful to the reviewer for recognizing and acknowledging our work. We will put the sentence "We observe... generality of order-preserving methods" in the main paper in Section 4.3.2. We will use "OP-non-TB methods" (instead of "non-OP-TB") to denote OP-FM, OP-DB, and OP-subTB methods for clarity. We will put a more in-depth comparison in the final version.

---

### Official Review · Reviewer_Ndf3 · 2023-10-30

**Soundness:** 3 good
**Presentation:** 3 good
**Contribution:** 3 good
**Rating:** 6
**Confidence:** 3

**Summary:**

The paper proposes OP-GFN, order-preserving GFlowNets, that sample with probabilities in proportion to a learned reward function instead of explicitly defined reward, the learned reward is compatible with a provided partial order on the candidates. OP-GFN promotes exploration in the early stages and exploitation in the later stages of the training by gradually sparsifying the reward function during the training. The authors provide theoretical proof that the learned reward is piecewise log-linear with respect to the ranking by the objective function. OP-GFN are proposed for both the single-objective maximization and multi-objective Pareto approximation problems, where experimental results on synthesis environment HyperGrid and two real-world applications NATS-Bench and molecular designs demonstrate competitive performance.

**Strengths:**

The theoretical contribution of the paper is sound and novel, where1)  the log reward being piecewise linear with respect to the ranking enables exploration on the early training stages and the exploitation on the later sparsified reward R is a useful and desirable feature in many training procedures; 2) matching the flow F with a sufficiently trained OP-GFN will assign high flow value on non-terminal states that on the trajectories ending in maximal reward candidates, enabling sampling optimal candidates exponentially more often than non-optimal candidates.

There is a range of comprehensive experimental results with details in the main document or in the supplementary material

The paper is well written and organized, making it easy to follow.

**Weaknesses:**

There can be more comparison against other state-of-the-art methods in the experimental section. The experiments, such as the hypergrid, only compares against the GFN-TB method. If the exploration-exploitation is the main benefit of OR-GFN in single objective problems, having more baseline comparison that encourages exploration of TB (such as https://arxiv.org/pdf/2306.17693.pdf or other variations of the reward besides beta-exponentiating) can make the paper stronger.

In single-objective maximization, the form of the local label y of the Pareto set is explicitly given in section 3.2, however in the multi-objective case, it is not clear how the label y can be calculated a priori, when the Pareto fronts are unknown. “When the true Pareto front is unknown, we use a discretization of the extreme faces of the objective space hypercube as P.” It is not clear whether the proposed method works due to the good estimate of the P, i.e., whether GGN can also benefit from incorporating estimated Pareto fronts into reward definition somehow.

**Questions:**

In section 4.2, the results shown are for the top-100 candidates, are the results similar for other choices of K?

In the experiment on neural architecture search, in Figure 4.2 the test accuracy comparison is against the clocktime, could you plot against the number of samples to understand the impact of learning the reward and sample efficiency of OR–GFN?

Have you considered other reward schemes besides the exponentially scaled reward? For example, some UCB-variant of the reward to balance between exploration vs exploitation?

Related work on encouraging exploration of GFlowNets: https://arxiv.org/pdf/2306.17693.pdf


Minor Comments:
Figure 4.1 caption, typo “Topk”
Figure 4.1, should the legend be ‘TB-OP’ instead of ‘TB-RP’?
In section 4.1, typo “TB‘s performance”

---

> ### Author Response · Authors · 2023-11-14
> **Response to Reviewer Ndf3**
>
> Thank you for thoroughly checking our paper, pointing out related work, and suggesting improvements to the experimental parts. We try to address your concerns below:
>
> 1. **Exploration in TB**. Thanks for your insightful reference. We will add the paper to the related work.  We think the idea of Thompson sampling could be useful in OP-GFNs as well. However, one of the main benefits of our work is that it naturally balances the exploration and exploitation as the training goes, without the need for manually designed exploration steps. Specifically, compared with Thompson Sampling, sampling from $K$ different forward policies increases the computational budget for $\times K$. Please see the **Common Response Related Work**.
>
> 2. **Pareto front** Sorry for the confusion, the sentence "When the true Pareto front is unknown, we use a discretization of the extreme faces of the objective space hypercube as $P$", is used only in the evaluation instead of the training. When we do not know the true Pareto front, we can only use evenly spaced points on the outermost faces of the $[0,1]^D$ cube to measure the closeness or diversity of the generated samples.
> In the training, we describe how to label the samples in "Section 3.1, 1) Order preserving". In every step, we sample a batch of terminal states $X$ and label 1 to those which are non-dominated in $X$, and 0 otherwise. We will add this description to make it more clearly in the paper. Please note the local distribution in Section 3.2 is a special case of those in Section 3.1.
>
> 3. **Value of $k$ in top$k$** We think it might be unnecessary to compare many different $k$s. Since $k$ is only the moving window to reduce the possible variance in measuring the optimal candidates.
>
> 4. **Plotting against the clock time** We follow Nats-Bench paper to plot against the clock time. We think when evaluating the accuracy is costly, it makes more sense to use the clock time than the number of samples to measure the sampling efficiency. When obtaining the reward is easy, such as in the molecular design, we then will use the number of samples.
>
> 5. **Reward schemes other than GFN-$\beta$**. Please see the **Common Response: Related Works**.
>
> 6. **Minor: typos**. Thanks for pointing out the typos. We have corrected them in the updated manuscript.

---

> > ### Comment · Reviewer_Ndf3 · 2023-11-22
> >
> > Thank you for addressing my questions and the additional experiments as well as discussion in the paper. I keep a positive score.

---

> > > ### Author Response · Authors · 2023-11-22
> > > **Response to Reviewer Ndf3**
> > >
> > > We are thankful to the reviewer for recognizing and acknowledging our work.

---

### Official Review · Reviewer_ipVV · 2023-10-30

**Soundness:** 3 good
**Presentation:** 3 good
**Contribution:** 3 good
**Rating:** 8
**Confidence:** 4

**Summary:**

The paper proposes to use the GFlowNet framework to learn samplers of optimal points based on given orderings. This is useful because it applies to single and multi-objective problems alike.

The method works by learning a reward function that minimizes the divergence between the optimality distribution (i.e. the indicator function around the argmax of $\mathcal{X}$) and the distribution induced by the learned reward function, as learned by a GFlowNet. This function is learned simultaneously with the corresponding GFlowNet sampler that tries to match this non-stationary reward, which at convergence neatly results in finding the optimal points.

**Strengths:**

The idea proposed in the paper is solid and the execution is well done; the method is tested on a whole variety of tasks and relevant setups.
The idea certainly relates to other rank-based methods, such as those in RL & search, but stands on its own in the GFlowNet framework.

**Weaknesses:**

My biggest criticism of the paper is really its presentation.

It's really not clear what the algorithm actually is, readers have to go all the way into the appendix to find it, and even there some questions remain, are $\mathcal{T}$ and $\mathcal{D}$ distinct? What is $\hat R$ trained on? Does it have distinct parameters? Shared? etc.
From scrolling through the appendix, it appears that there are, understandably, a number of tricks that can be used. Most of them have some form of ablation in one task or the other, but a cohesive summary is lacking. The main criticism here is that it's important to disentangle what the contribution is from the algorithm itself, and from the tricks.

Investigative figures like Fig. E.6 should really be in the main body of the paper, and it would be much nicer to have such a figure for more complex domains. Generally the paper doesn't do a great job of showing _why_ the method works empirically.

**Questions:**

Notes:
- For someone not familiar with GFNs, Eq (3) may not be obvious (R being parameterized "through" P_F P_B & Z)
- I'm not sure that I fully appreciate the "piecewise linear" part of the contribution. Is there something I'm missing?
    - First of all, the whole thing is discrete and operating on sets. It makes no sense to talk about piecewise linearity because we are not in $\mathbb{R}$, the pieces "in between" don't even exist.
    - Ignoring the above, the shape of the function doesn't really matter? What matters is $\mathrm{order}(a,b) \equiv \mathrm{order}(\hat R(a), \hat R(b))$, which follows in a fairly straightforward way once we assume 0 loss.
- Proposition 3/6 is similarly weird, why this specific choice of MDP? I'd recommend either introducing why this choice matters, or maybe at least state that the general case is much harder to provide theoretical statements for. I find the current form a bit awkward
- 4.2; the authors seem to be using the molecular setup of Shen et al., not Bengio et al.. Shen et al.'s setup is quite different (and more simple, the state spaces are made to be much small so that computing $p(x; \theta)$ becomes tractable). It would be appropriate to note this. It's not clear what "choices" the authors are using from Shen et al. since quite a few are introduced in that work (which appear to have a significant impact)
- "optionally use the backward KL regularization (-KL) and trajectories augmentation (-AUG) introduced in Appendix E.1", these options don't seem to have that much impact, but it's a bit weird that they're just mentioned in passing in the main text.
- "we observe that OP-GFNs can learn a highly sparse reward function that concentrates on the true Pareto solutions, outperforming PC-GFNs." It's a bit weird to claim here that OP-GFNs are outperforming PC-GFNs, because they learn entirely different things. From Fig 5.1 it seems like PC-GFN is attributing some probability mass on the Pareto front points, which suggests to me that it has discovered them.
    - This is maybe more of a comment on RL literature, but I'm also not a fan of comparing algorithms on a grid. Gridworlds are useful to sanity check algorithms, not to compare algorithms.
- In Sections 4 & 5, it would be good to clarify what work introduced what task and what MDP setup which is being used.
- Why build on GFlowNets? Don't get me wrong I'm a big fan of GFNs, but it seems to me like the use-case of that framework is to obtain "smooth" energy-based amortized samplers, whereas the proposed work is meant to converge to an extremely peaky distribution where essentially $p(x^*)=1$, and is much more akin to finding the optimal greedy policy in RL. There may be a very good reasons (I can think of some ;) but I think the paper could do a better job of explaining them.

---

> ### Author Response · Authors · 2023-11-14
> **Response to Reviewer ipVV**
>
> Thank you for thoroughly checking our paper and for providing valuable feedback about notations, structure and content presentation. However, with the current paper layout and the given page limit, we see ourselves faced with a lot of restrictions, that make it very hard to restructure the whole paper. Addressing your proposals one-by-one:
>
> **Weaknesses**:
>
> 1. **Minor: $\mathcal{D}$ or $\mathcal{T}$**. Sorry for the confusion, $\mathcal{D}$ is a typo, it should indeed be $\mathcal{T}$ in Algorithm 1. We have corrected it in the updated manuscript.
>
> 2. **What $\widehat{R}$ is trained on?** $\widehat{R}$ is trained on the order-preserving loss $L_{OP} $, its parameters depend on GFN parametrization used, i.e., the parameters of: 1) $P_F$, $P_B$ and $Z$, 2) $P_F$, $P_B$ and $F$, or 3) $F$, as given in Table 2.1. parametrized by the GFN parametrization, such as $\widehat{R}_{\rm TB}(x;\theta)$.
>
> 3. **Disentangling algorithms with tricks**. Sorry for the confusion. We optionally use backward augmentation and backward KL regularization in the NAS environments, and clearly specify their usage by (-KL) and (-AUG) in the plots or tables. We have observed from Figure 4.2 that the major improvements are from the OP method, and these tricks only yield marginal improvements. More specifically, all 3 variations of OP-GFNs that include selections of KL regularization or training trajectory augmentation are very close to each other. In other environments, we did not include additional tricks when comparing with traditional GFNs but followed the same setup as the original papers. In other words, the only difference is the training criteria.
>
> 4. **Investigative figures in the appendix**. We admit some figures are left in the appendix. However, there is a 9-page limit, so we chose to put the ablation studies and auxiliary results in the appendix. We felt, that discussing the properties of the trained OP-GFN gives a good insight about what kind of generating distribution the OP-GFN learns. Hence, we sacrificed some space to include these theoretic results in the main paper and put the validation of the theoretical results in the appendix.
>
> **Questions**:
>
> 1. **Descriptions of GFNs**. We succinctly describe the GFNs in Table 2.1. We have made a clearer reference to this table in the updated manuscript.
>
> 2. **Log piecewise linear**. We admit "log piecewise linear"  is not a good naming, and might be a confusing term. Following your arguments, we replace it with “piecewise geometric progression”. The result of "piecewise geometric progression" is not as trivial as it sounds, especially when there are candidates of the same objective value $u(\cdot)$, please refer Propositions 2 and 5 for details.
> % \lukas{However, even if it is trivial it characterizes the distribution that is learned by the OP GFN. Therefore, we feel it is a necessary part of the paper in order to be complete.}
>
> 3. **Sequence prepend/append MDP**. Sequence prepend/append MDP are widely used in molecule generation, such as in Section 4.2, “For the SIX6, PHO4, QM9, sEH environment, we use the sequence prepend/append MDP in Definition 1”. Therefore, theoretical analysis based on this MDP structure is meaningful.
>
> 4. **Molecular setup**. We inherit the basic setup as Shen et al for trajectory balance, and for OP methods, we only change the training loss. Please see Section E.3.2, and **Common Response: Extensive Experiments**.
>
> 5. **Definition of -KL and -AUG**. We write the names in the main text because it appear in Figure 4.2, we defer the detailed description to the appendix due to the page limit.
>
> 6. **PC-GFNs vs OP-GFNs in HyperGrid**. In Figure 5.1, PC-GFNs indeed attribute some probability mass on the Pareto front points, but not on all such points, such as "currin-shubert", it misses a lot of Pareto front solutions. Please see Figure F.1, this is mainly because "currin-shubert"'s Pareto front is concave, and linear scalarization in PC-GFNs inherently cannot deal with a concave Pareto front. We point out this weakness in Section 2.2. On the contrary, OP-GFNs can recover all the solutions, regardless of the Pareto front shape, and the landscape terminal flow is much more sparse than PC-GFNs.
>
> 7. **Sanity checking in HyperGrid**. Thanks for your advice on sanity checking in the grid-worlds. We perform the sanity checking in HyperGrid in the multi-objective optimization, since only in the hypergrid, the Pareto front can be explicitly calculated. We also perform extensive experiments on real-world environments, including molecular designs and NAS.
>
> 8. **Details in Section 4, 5**. We put the detailed setup in Sections 4 and 5 in the appendix due to the page limit. We hope the experiments' succinct description and reference to the appendix in the main text, as well as "Appendix A: an overview" could help you navigate the detailed settings.
>
> 9. **RL vs OP-GFNs**. Please see the **Common Response: RL vs OP-GFNs**.

---

> > ### Comment · Reviewer_ipVV · 2023-11-22
> >
> > Thanks for the response and clarifications. I think the method is great, and am maintaining my score. I encourage the authors to keep working on the presentation and writing to make this work as impactful as possible.
> >
> > Note that, for molecules, I pointed out the use of Shen et al.'s setup because I thought it should be clearer what scale of problems are being tackled in this paper. Many GFlowNet papers have used the sEH reward with a fragment-graph-based MDP with ~100 fragments, which is _much_ harder (graphs rather than sequences are generated, and $|X|$ is at least $10^{16}$) than the append-prepend MDP with ~20 fragments ($|X|$ is about $10^7$).

---

> ### Author Response · Authors · 2023-11-23
> **Response to Reviewer ipVV**
>
> We are thankful to the reviewer for recognizing and acknowledging our work. We will continue working on improving our writing and presentation skills.
>
> We are also grateful for the identification of various sEH setups in different papers. This will be emphasized in Section 4.2 and Appendix E.3 in the updated paper. Besides, we have utilized the fragment-graph-based MDP in "Section 5.4: Fragment-Based Molecule Generation" to analyze OP-GFN's multiobjective performance, where maximizing sEH is one of the objectives.

---

### Official Review · Reviewer_8SYj · 2023-11-01

**Soundness:** 3 good
**Presentation:** 3 good
**Contribution:** 3 good
**Rating:** 6
**Confidence:** 3

**Summary:**

The paper introduces an innovative approach for training GFlowNets, eliminating the necessity for an explicit formulation of the reward function while ensuring compatibility with the provided order of candidates. This method involves the simultaneous training of a reward function, which maintains the order of samples in conjunction with GFlowNet. The authors conducted comprehensive experiments in both single and multi-objective settings, revealing that their proposed approach yields outstanding results in comparison to prior methods when dealing with only pairwise order relations and non-convex Pareto fronts.

**Strengths:**

The paper conducts extensive experiments under different objective settings and domains. Especially, the paper conducts experiments on NAS benchmark, which is a first attempt to apply GFlowNets into NAS while it is natural as we can make neural architecture by adding operations in a sequence manner. It also achieves superior results compared to other baselines in NAS benchmark.

The paper also tackles multi-objective problems with a non-convex Pareto front, which is hard to solve with prior multi-objective GFN methods such as PC-GFN. Additionally, the paper eliminates the necessity for fine-tuning the temperature parameter, β, which plays a critical role in GFlowNets' performance.

**Weaknesses:**

1. There is a possibility of encountering non-stationarity issues when jointly training GFlowNets and the reward function. It might be worth exploring alternative training strategies to mitigate this potential challenge.

2. Experimental results are not that persuasive, having little improvement over baselines. For example, this work just compares with simple GFN baseline in molecular tasks, more competitive baselines (e.g., subTB, FL-GFN, RL methods) are needed.

---


**Discussion needed regarding temperature conditioning methods**

There are some researches on temperature-conditioned GFlowNets, which learn a single model for multiple reward functions conditioned on different temperatures. It may be more persuasive to compare the proposed method with the following literature [1], [2], especially in single-objective settings. Note that reviewer understands that temperature-conditioned GFlowNets are too recently proposed, so the authors could not reflect this in this submission but suggest making some discussion in the final version for future researchers!

[1] Zhang, David W., et al. "Robust scheduling with GFlowNets." arXiv preprint arXiv:2302.05446 (2023).

[2] Kim, Minsu, et al. "Learning to Scale Logits for Temperature-Conditional GFlowNets." arXiv preprint arXiv:2310.02823 (2023).

---

**Decision**

This work is novel, well-written, easy to understand, and provides insight for GFN communities. Although experimental results do not strictly outperform every competitive baseline, this work provides a lot of experiments over various tasks. To this end, the score would be 6.

**Questions:**

**Section 4.2**

Authors say that they compare the proposed method with GFN-$\beta$, where $\beta$ is the selected value from the previous work. As far as I know, the temperature is not tuned in the previous work. What if we assign high values to $\beta$? It seems that we can achieve higher topk reward by sacrificing diversity when we assign high values to $\beta$ and authors only consider maximal objective in single-experiment setting

**Section 5.2, 5.4**

Authors say that they use a replay buffer and do off-policy training. I am curious that as the reward function is trained across training, off-policy training may lead to degrading performance.

**Minor Questions**

Section 4.1) $u_0\$ seems a typo. Maybe $R_0$?

Section 4.3) Authors propose two implementation tricks, KL regularization and trajectories augmentation. Especially trajectories augmentation has shown superior results in the previous paper. It is hard to capture when those two tricks are applied. While authors say trajectory augmentation is applied in molecule design (E.3.2), main paper say that trajectories augmentation is optionally used in NAS environment. It may be helpful when those two tricks are applied across differnt experiment settings.

---

> ### Author Response · Authors · 2023-11-14
> **Response to Reviewer 8SYj**
>
> Thank you for your well-structured feedback and for pointing out very recent, and strongly related literature. As mentioned in the general response, we added a discussion to the related work section. Concerning your questions:
>
> 1. **Non-stationarity issues**. We find the non-stationary issues are not very serious in the single-objective case. On the contrary, if we do not learn but define a reward in prior, like with GFN-$\beta$, we cannot obtain similar performance like with OP-GFNs. defining a reward in prior by GFN-$\beta$ methods are not as good as the OP-GFNs. Please see Appendix E.4.4. ablation studies. However,  In the multi-objective cases, we observe the non-stationary issues arise, and we choose to use the replay buffer to stabilize the training. Please see the second paragraph of Section 5.2. We added a sentence to make this more clear.
>
> 2. **More competitive baselines** From Figure 4.1, Figure 4.2, Figure 5.1, Figure F.1, our algorithm has proven to be more effective than previous methods. We have already include the subTB (Appendix E.4.5) and RL methods (REINFORCE, Figure 4.2) in NAS environments, which are, to our knowledge, strong baselines. Could you please elaborate in more detail, which baselines you would like to see in comparison? For the molecular design, please see the **Common Response: Extensive Experiments**.
>
> 3. **Temperature conditioning methods** Please see the **Common Response: Related Work**.
>
> 4. **Setting higher $\beta$** Using GFN for stochastic optimization, (1) we do not only consider the maximal reward, but we also need to pay attention to the diversity (exploration). For example, in Section 4.2 (molecular design) we also measure the number of different optimal candidates. If the reward is very sparse, the sampler will only visit states with near-zero rewards, and cannot explore further to visit the various different maximal candidates. We validate this in Figure E.4, when setting $R_0=0$, it yields a very sparse reward landscape. As a result, and the sampler can only sample candidates near the initial points. (2) even if we only consider the maximal value regardless of the diversity, a large $\beta$ will still not be a good choice. Since a large $\beta$ might hinder the exploration to optimal candidates distant to the initialization. To point out this problem, we tested various For example, we test various $\beta$ in NAS in Figure E.8. Setting $\beta=128$ does not perform better than $\beta=16$.
>
> 5. **Replay buffer** In the sparse reward landscape, off-policy training has been shown to help stabilize the training. Please refer to Appendix C.1 of the paper "Goal-conditioned GFlowNets for Controllable Multi-Objective Molecular Design" (https://arxiv.org/pdf/2306.04620.pdf). We add an additional sentence in Section 5.2 to state it more clearly.
>
> 6. **Minor: $u_0$ or $R_0$** We always use $R_0$ to denote the constant in the objective of HyperGrid in the whole paper. Since either $u_0$ or $R_0$ merely represents a constant, we decide to use $R_0$ to align with the whole paper's notations, including the updated Figure E.5.
>
> 7. **Minor: trajectory augmentation** We always use trajectory augmentation in molecule design as suggested in Appendix E.3.2. In the NAS environment, we optionally use trajectory augmentation, and use "-AUG" to if we use this augmentation in the algorithm, see Figure 4.2.

---

> > ### Comment · Reviewer_8SYj · 2023-11-20
> > **Nice work**
> >
> > Thank you for addressing my concerns with your responses. They have provided clarity, and I maintain a positive score.

---

> > > ### Author Response · Authors · 2023-11-22
> > > **Response to the reviewer 8SYj**
> > >
> > > We are thankful to the reviewer for appreciating our work.

---

### Official Review · Reviewer_BMH3 · 2023-11-01

**Soundness:** 3 good
**Presentation:** 3 good
**Contribution:** 3 good
**Rating:** 6
**Confidence:** 4

**Summary:**

This work proposes to extend GFlowNets, called Order-Preserving GFlowNets (OP-GFN), to sample candidates in proportion to a reward function such that the sampling is consistent with the provided partial order of the candidates. By this extension, they show how exploration and exploitation can be controlled, such that candidates higher in the hierarchy can be given more preference. The method shows benefits for both single objective and multi-objective cases by conducing experiments on a variety of tasks.

**Strengths:**

1. The paper introduces an important extension of the GFlowNets for multi-objective optimization when D > 1 objectives need to be optimized.
2. The work also discusses how an efficient utilization of the GFlowNet policy can be achieved in difficult to explore settings.
3. The theoretical results and analysis are useful to understand the proposed method and its advantages.
4. The work also provides a good overview of the literature to benefit the reader.

**Weaknesses:**

1. The experiments section can be expanded to include more difficult environments. For example, for hypergrid,higher values of H and N can be tested as larger grids will help analyzing the exploration problems better.
2. Detailed balance objective can perform reasonably well in many settings. It will be beneficial to include it in all methods and numbers reported.
3. It will be useful to add standard deviation and error bars across experiments. It will also be useful to better understand the variance across different GFlowNet objectives and baseline methods.

By addressing these concerns with experiments, the authors will help address the empirical limitations to strengthen their theoretical discussions and the overall contribution.

**Questions:**

1. Is it possible to include some more difficult configurations of the environments? Hypergrid is one easy example for such an experiment.
2. It would be helpful if the variations across different seeds and runs could be provided for the experiments.
3. A hyperparam search for the best learning rate can also be sometimes a big contributing factor across different learning methods.

If these could be provided along with the comments in the weaknesses, section it will help in addressing most of my concerns towards the experimental contribution of this work.

---

> ### Author Response · Authors · 2023-11-14
> **Response to Reviewer BMH3**
>
> Thank you for reviewing our paper and for providing a specific list how to improve our work. We agree with your assessment and took the effort to run additional experiments, in the hope of raising the score. We address your concerns below:
>
> **1. More complex environments**. In the submitted paper, we included many complex real-world environments apart from apart from HyperGrid, which includes molecular generation tasks and NAS. Such tasks are standard in evaluating GFNs in previous papers. We mostly use HyperGrid only for sanity-checking and therefore did not consider more complex configurations with larger $H$ and $D$ at first. To understand the exploration of the algorithm better, in the molecular design environments, we record the number of distinct optimal candidates. Please see the **Common Response: Extensive Experiments**. Furthermore, we additionally consider two high-dimensional settings in HyperGrid: $(D,H)=(3,32)$ and $(D,H)=(4,16)$, please see Section 4.1, and updated Figure E.5. We consider the following three ratios to measure exploration-exploitation: 1) \#(distinctly visited states)/\#(all the states); 2) \#(distinctly visited maximal states)/ \#(all the maximal states); 3) In the most recently 4000 visited states, \#(distinctly maximal states)/4000. A good sampling algorithm should have a small ratio 1), and a large ratio 2), 3).  We confirm that TB’s performance is sensitive to the choice of $R_0$, and observe that OP-TB can recover all maximal areas more efficiently, and sample maximal candidates with higher probability after visiting fewer distinct candidates.
>
> **2. Including the detailed balance objective.** The submitted paper contains many experiments with detailed balance in the NAS environment in Appendix E.4.5, where we do not observe a significant gain from the trajectory balance. Therefore, we did not include detailed balance in other environments. However, we ran additional experiments and added detailed balance to the molecular design. Please see the **Common Response: Extensive Experiments**.
>
> **3. Adding error bars**. We agree with you and add the error bars to the experiments in the main paper, i.e., to the molecular design environments (Please see the **Common Response: Extensive Experiments**). We observe that the OP-TB's variance is small, and outperforms other methods beyond the error bars (especially in sehstr).
>
> **4. A hyperparameter search for the best learning rate**  We use the Adam optimizer for training criteria and parametrizations (DB, TB, subTB, OP). The Adam optimizer adaptively computes individual learning rates for
> different parameters. Therefore, we did not finetune the learning rate of the Adam optimizer. For example, in the NAS environment, we use the initial learning rate $10^{-3}$ for network parameters and $10^{-1}$ for $Z$'s parameters.
> We think this comparison is fair because it shows that OP-GFN can outperform standard GFN by only replacing the training criterion. In other words, our results are not cherry-picked from the learning rate.

---

> ### Author Response · Authors · 2023-11-22
> **Any further comments from the Reviewer BMH3?**
>
> We appreciate your time and effort in reviewing our paper. Given the upcoming deadline for the end of the discussion period, we are wondering whether the added experiments and our responses have adequately addressed your concerns. We are willing to make more improvements based on your further comments.

---

> ### Comment · Reviewer_BMH3 · 2023-11-22
>
> Thank you for clarifying the concerns and for updating the work. I have revised my score.

---

> ### Author Response · Authors · 2023-11-22
> **Response to Reviewer BMH3**
>
> We are thankful to the reviewer for recognizing and acknowledging our work.

---

### Author Response · Authors · 2023-11-14
**Common Response**

Thank all reviewers for your insightful feedback. We used your comments to update our manuscript. All changes to the submitted paper version are highlighted in red. There are some common issues that are raised by a number of reviewers, that we want to address first:

**1. Contributions**: We summarize our contributions here. We propose the order-preserving GFNs, that (1) **balance the exploration-exploitation** in the training without requiring prior distribution on the temperature; (2) do not require a scalar target reward function, only use the (partial)-ordering relation, and **well-designed for both the single and multi-objective optimization problems**; (3) achieve **better performances** than previous GFN and (RL-)sampling methods in terms of diversity and top performance.

**2. Related work**: We thank the reviewers for pointing out the very relevant papers [1], [2], [3]. We will discuss it in the related work section of our updated manuscript.

Paper [1] demonstrates how Thompson sampling with GFlowNets allows for improved exploration and optimization efficiency in GFlowNets. We point out that the key difference in the exploration strategy is that [2] encourages the exploration by bootstrapping $K$ different policies, while in the early stages of OP-GFNs, the learned reward is almost uniform, and thus the forward policies are close to random, which encourages the exploration.  As the training goes on, the learned reward gets sparser, the exploration declines and exploitation arises. We remark that the idea of Thompson sampling can be used in the exploitation stages of OP-GFNs to further encourage the exploration in the latter stages of training.

[2,3] propose the Temperature-Conditional GFlowNets (TC-GFNs) to learn the generative distribution with any given temperature. [2] conditions the policy networks on the temperature, and [3] directly scales probability logits regarding the temperature values. We point out some critical differences to OP-GFN: 1) In TC-GFNs, a suited $\beta$'s prior must still be chosen. while OP-GFNs do not require such a choice. 2) TC-GFNs learn to match  $R^{\beta}(x)$ for all $\beta$, while OP-GFNs just learn to sample with the correct ordering statistics. However, using these few statistics, OP-GFNs still achieve competitive results in both single and multi-objective optimization. 3) TC-GFNs require the scalar reward function, while OP-GFNs can be directly used in multi-objective optimization.

We have implemented the temperature-conditioning GFNs in multi-objective optimization, where we encode the preference vector together with temperature in preference-conditioning GFNs. For example, in HyperGrid in Section 5.1, when training the PC-GFNs, we sample $\beta\sim \Gamma(16,1)$, and sample with $\beta=16$ in evaluation, in Figure 5.1 and metrics table. We observe that the learned reward cannot recover all the Pareto front solutions, as shown in the second row (OP-GFNs' results).

[1] Zhang, David W., et al. "Robust scheduling with GFlowNets." arXiv preprint arXiv:2302.05446 (2023).

[2] Kim, Minsu, Joohwan Ko, Dinghuai Zhang, Ling Pan, Taeyoung Yun, Woochang Kim, Jinkyoo Park, and Yoshua Bengio. "Learning to Scale Logits for Temperature-Conditional GFlowNets." arXiv preprint arXiv:2310.02823 (2023).

[3] Zhang, David W., Corrado Rainone, Markus Peschl, and Roberto Bondesan. "Robust Scheduling with GFlowNets." In The Eleventh International Conference on Learning Representations. 2022.

**3. RL vs OP-GFNs** We agree, that the link to methods was not well pointed out in the submitted paper. Regarding your questions about RL and OP-GFNs: We thank the reviewers for their insightful comments. Regarding the questions of RL and OP-GFNs, (1) OP-GFNs are designed for both single and multi-objective optimization. Sampling optimal candidates in the single objective optimization is only part of the OP-GFNs' ability. OP-GFNs can approximate the Pareto front, while RL cannot be directly used in this problem. (2) GFNs can sample more diverse modes than RL, which has been verified by previous applications of GFNs. From the experiments in molecular design in **Common response: Extensive Experiments**, we observe that OP-GFNs indeed sample more diverse optimal candidates than RL baselines.

---

### Author Response · Authors · 2023-11-14
**Common Response: Extensive Experiments**

We thank the reviewers for pointing out weaknesses in our experimental setup. their insightful comments. We have performed extensive experiments on NAS in the ablation studies (Appendix E.4.4 and Appendix E.4.5)  in the submitted paper version. We have added a clearer reference in Section 4.3.2. Furthermore, we include two more sets of experiments in the updated manuscripts.

- We updated the HyperGrid experiments with higher dimensional settings and considered three ratios to measure exploration-exploitation in Section 4.1.  We set $(D,H)=(2,64)$, $(3,32)$ and $(4,16)$, and compare TB and order-preserving TB (OP-TB). We additionally plot the following three ratios: 1) \#(distinctly visited states)/\#(all the states); 2) \#(distinctly visited maximal states)/ \#(all the maximal states); 3) In the most recently 4000 visited states, \#(distinctly maximal states)/4000. A good sampling algorithm should have small ratio 1), and large ratio 2), 3), which means it can sample diverse maximal states (large ratio 2), exploration), and sample only maximal states (large ratio 3), exploitation), using the fewest distinct visited states (small ratio 1), efficiency). We observe from [this figure](https://i.imgur.com/ewVK2xq.jpg) that OP-TB **outperforms TB in almost every $R_0$ in terms of three ratios**.

- We decided to add more baselines to the molecular design environments, including 1 Markov sampling algorithm (mars), 3 RL algorithms (a2c, ppo, sql) and 5 GFN algorithms (tb, db, subtb, maxent, sub). We run our experiments starting from 3 seeds and report the mean and variances. We report the Top-K reward and number of optimal candidates to measure the exploitation and exploration respectively. Please see Section 4.2 for details. In particular, we updated Figure 4.1 to the updated manuscript, including previous [GFN methods](https://i.imgur.com/egvgnlk.jpg) and [RL methods](https://i.imgur.com/bDXKQCG.jpg). We find that the order-preserving method can **outperform all the baselines in both the ability to find the number of different optimal candidates and the average top-$k$ performance**, especially in the diversity of the solutions. We update the experimental settings in Appendix E.3.

We hope, that these additional experiments could increase the merit and value of our paper.

---

### Comment · Area_Chair_1EFA · 2023-11-22
**Discussion period ending today**

Dear reviewers,

This a reminder that deadline of author/reviewer discussion is AOE Nov 22nd (today). Please engage in the discussion and make potential adjustments to the rating and reviews.

Thank you!
AC

---

### Meta-Review · Area_Chair_1EFA · 2023-12-04

**Metareview:**

The authors proposed Order-Preserving GFlowNets which can be used to balance explore-exploit for single and multi-objective optimization. The experiments have shown good performance for real-world tasks including NAS and molecular design. The theoretical results are interesting and useful. The proposed method and analyses are novel and have potential impact on the black-box function optimization community. The reviewers reached consensus to accept this paper.

The reviewers have also suggested further revisions to incorporate in-depth comparison results on OP-TB and other baselines, and polish the presentation. Please incorporate those feedback in the final version.

I recommend to accept this paper.

**Justification For Why Not Higher Score:**

The presentation of this work needs to be further improved to allow more impact on the research field. While the authors have added more experiments on complex environments, the method does not make a super significant improvement over baselines.

**Justification For Why Not Lower Score:**

The experiments have shown good performance for real-world tasks including NAS and molecular design. The theoretical results are interesting and useful. The proposed method and analyses are novel and have potential impact on the black-box function optimization community.

---

### Decision · Program_Chairs · 2024-01-16

Accept (poster)